# Molecular sociology of virus-induced cellular condensates supporting reovirus assembly and replication

Xiaoyu Liu [1,2,6], Xian Xia [1,2,6], Michael W. Martynowycz[3,4,5],
Tamir Gonen [3,4] & Z. Hong Zhou [1,2]

Virus-induced cellular condensates, or viral factories, are poorly understood high-density phases where replication of many viruses occurs. Here, by cryogenic electron tomography (cryoET) of focused ion beam (FIB) milling-produced lamellae of mammalian reovirus (MRV)-infected cells, we visualized the molecular organization and interplay (i.e., "molecular sociology") of host and virus in 3D at two time points post-infection, enabling a detailed description of these condensates and a mechanistic understanding of MRV replication within them. Expanding over time, the condensate fashions host ribosomes at its periphery, and host microtubules, lipid membranes, and viral molecules in its interior, forming a 3D architecture that supports the dynamic processes of viral genome replication and capsid assembly. A total of six MRV assembly intermediates are identified inside the condensate: star core, empty and genome-containing cores, empty and full virions, and outer shell particle. Except for star core, these intermediates are visualized at atomic resolution by cryogenic electron microscopy (cryoEM) of cellular extracts. The temporal sequence and spatial rearrangement among these viral intermediates choreograph the viral life cycle within the condensates. Together, the molecular sociology of MRV-induced cellular condensate highlights the functional advantage of transient enrichment of molecules at the right location and time for viral replication.

While membrane-bound organelles provide a barrier against entropy flow for cellular energy generation[1], membrane-less cellular condensates are key to information preservation, playing pivotal roles in DNA replication, RNA transcription, and protein synthesis[2,3]. Cellular condensates are molecule-dense regions distinctive from their surroundings in a cell without the use of membrane. Some are also referred to as liquid-like phases, granules, and in the case of viral replication or assembly in host cells, viral factories. Such condensates selectively enrich specific molecules and facilitate biological reactions[4,5]. Cellular condensates are involved in various biological processes, including cell signaling and response[6,7], energy and biomolecule storage[8], and development of Parkinson's and Alzheimer's diseases[9], underscoring their significance in both maintaining cellular homeostasis in health and causing tissue malfunction in diseases. Recent studies have also revealed the important role of cellular condensates in the life cycles of many viruses[10,11]. Viruses organize their genome replication, transcription, and capsid assembly within an organelle-like condensate[12], also known as viral factory[13,14], to both

[1]Department of Microbiology, Immunology and Molecular Genetics, University of California, Los Angeles, CA, USA. [2]California NanoSystems Institute, University of California, Los Angeles, CA, USA. [3]Howard Hughes Medical Institute, University of California, Los Angeles, CA, USA. [4]Department of Biological Chemistry, University of California, Los Angeles, CA, USA. [5]Present address: Hauptman-Woodward Medical Research Institute, Buffalo, NY, USA. [6]These authors contributed equally: Xiaoyu Liu, Xian Xia. ✉e-mail: Hong.Zhou@UCLA.edu

promote replication efficiency and shield viral intermediates from host defenses[11,15]. Fluorescence microscopy reveals that viral condensates appear as puncta and demonstrate their liquid-like properties, including the ability to grow and fuse[16–18]. Growing evidence shows that liquid–liquid phase separation mediates the formation of certain viral condensates[19,20]. However, the detailed 3D organization of these condensates within the cellular environment, and their role in facilitating complex viral replication processes, remain largely unknown. In this study, we employed an integrated approach of cellular cryogenic electron tomography (cryoET) and high-resolution cryogenic electron microscopy (cryoEM) to study the architecture and mechanisms of mammalian reovirus (MRV)-induced cellular condensates.

MRV is a prototypical member of the *Reoviridae* family of segmented double-stranded RNA (dsRNA) viruses, which is further divided into two subfamilies based on the presence or absence of turrets on the inner capsids: turreted (*Spinareovirinae*, including MRV) and non-turreted (*Sedoreovirinae*, including bluetongue virus (BTV) and the life-threatening rotavirus)[21,22]. MRV capsids are composed of two concentric protein layers, encapsulating 10 dsRNA segments. The outer layer of MRV consists of the protection protein σ3, membrane penetration protein μ1, and spike protein σ1. The inner layer consists of proteins λ1, σ2, and turret protein λ2[23]. The transcription enzyme complex (TEC) is comprised of λ3 and μ2 and is attached inside the inner layer[24]. After entering the cell, the virion undergoes a series of steps, including the uncoating of its outer layer proteins, becoming first the infectious subviral particle (ISVP), then the core particle[25]. Release of the core particle from the endosome activates mRNA transcription[26–28]; subsequent RNA transcription, genome replication, and progeny virus assembly all occur within MRV-induced condensates. The nonstructural protein μNS forms the scaffold of viral factories[29,30], which can be considered as dynamic liquid-like condensates[5,13,19]. In contrast to the uncoating process from virion to core, the MRV genomic RNA packaging and virus assembly pathway remain elusive, with limited intermediates identified and structures solved[31,32]. Moreover, within the condensates, there may be additional transient states.

In this work, in order to explore the architecture of the condensate and the complex activities of MRV within it, we investigated the morphology of MRV-induced cellular condensates and the changes in host cells at two time points post-infection. This study reveals the intricate interplay between viral components and host factors, offering a molecular sociology perspective. By capturing various intermediates, we elucidate the mechanism of MRV assembly and genome packaging, regulated within the condensates.

## Results
### 3D architecture of cellular condensate for MRV assembly
To directly visualize the in situ spatiotemporal organization (i.e., molecular sociology) of the virus-induced cellular condensate, we carried out cryoET of MRV-infected cells at two post-infection time points: 6 h post-infection (early infection stage) and 48 h post-infection (late infection stage) (Fig. 1a). Low-magnification cryoEM images of cell lamellae at both time points clearly show that MRV viral factories are cytoplasmic, non-membrane-bound condensates (Figs. 1b,g and S1a,b). At 6 h post-infection, multiple small viral condensates appear in dispersed spots near host cellular machineries (Figs. 1b and S1a). In contrast, at 48 h post-infection, one large condensate appears as a para-crystalline array of viral particles, with no cellular structures inside except for microtubules (Figs. 1g and S1b). Live-cell microscopy has revealed dynamic fusion events for MRV condensates in which small condensates merge with larger ones[13]. The large condensate we observed at 48 h post-infection is likely the result of such fusion events. With higher-magnification cellular cryoET tomograms, we were able to visualize these condensates in greater detail, allowing us to explore the various viral and cellular structures and interaction networks that govern MRV replication and assembly.

The tomograms resolve many cellular structures around or within these condensates, including microtubules, ribosomes, host lipid membranes, mitochondria, Golgi, vesicles, and the nucleus (Figs. 1c–f, h and S1c-g; Movie S1, S2). Microtubules are present within the condensates at both timepoints. These microtubules appear either as undecorated filaments, with a diameter of ~24 nm (Fig. S1d), or as decorated filaments, with a diameter of ~50 nm (Fig. S1e). Dynein, a microtubule motor protein, has been localized in MRV induced condensates and shown to be required for forming large condensates[29,33]. Whether the microtubule-decorating densities (Fig. S1e) are dynein motors or not awaits further investigation. Previous studies show that depolymerization of microtubules by nocodazole inhibited the merging of small condensates into larger ones and lowered genome packaging efficiency[34], highlighting the important role of microtubules in MRV-induced condensates[13,33]. Ribosomes appear in the vicinity of the condensates (Fig. S1f, g), likely for viral protein synthesis. This observation aligns with earlier immunofluorescent results showing that active translation of reovirus mRNA occurs within MRV viral factories, and that the translational machinery, including ribosomal subunits and cellular translation factors, localizes to viral factories[35]. Additionally, lipid membranes from host cells are observed to be involved in the condensates and interact with virus particles (Fig. S1c). Endoplasmic reticulum membranes were reported to be remodeled by reovirus non-structural protein σNS or μNS into tubular or vesicular forms, participating in the formation of MRV viral factories/condensates[36]. Lipid droplets have been reported to play a role in the regulation and assembly of outer capsid proteins[37]. Taken together with these previous results, our cellular cryoET data reveal the 3D architecture of MRV-induced cellular condensates, their morphological transitions over time, and their hijacking of the host machinery for virus replication, assembly, and transport.

### Viral assembly intermediates within the condensate
To thoroughly understand the MRV assembly process as it occurs within cells, we analyzed the tomograms, focusing on viral assembly intermediates and products (Fig. 2a, b; Movie S3). MRV virions consist of two concentric protein layers, encapsulating 10 dsRNA segments (Fig. 2c). Both single-layered (inner shell only) and double-layered particles, with and without genomic RNA inside, were observed in the condensates (Fig. 2a,d). Through subtomogram averaging of cellular tomograms from 6 h and 48 h post-infection, we obtained six icosahedral reconstructions of MRV assembly states (i.e., star core, core, virion, empty core, empty virion, and outer shell particle) from each time point (Figs. 2f and S2). The single-layered particles (star core, core, and empty core), comprising only inner-layer proteins, represent the early assembly states of MRV, while the virion and empty virion are the mature products. Meanwhile, the outer shell particle, consisting of the outer layer proteins μ1, σ3, and λ2, appears to be a byproduct of MRV assembly. Viral intermediates and products, including core and virion, concentrate within the same condensate, gathering in a disorderly manner without a clear patterned spatial distribution (Figs. 1, 2a and S1).

The cellular tomograms reveal single-layered particles (SLP) with different levels of invagination at the fivefold axis: star-shaped, hexagon-shaped, and spherical particles (Fig. 2d). The internal density within spherical SLP appears identical to that of the virion particle (Fig. 2d), indicating that dsRNA is fully packaged in the spherical SLP. In contrast, the densities inside star-shaped and hexagon-shaped SLPs are much lower than the dsRNA densities visualized in virion particles, suggesting that only ssRNA or no RNA is present inside. This shape transformation suggests that the particle expands during genome packaging and RNA replication. Core particles coated to various completeness degrees with outer layer proteins μ1 and σ3 were identified (Fig. 2d, red arrows), illustrating a sequential assembly of the MRV outer layer to finalize the mature virion particles: turret protein λ2

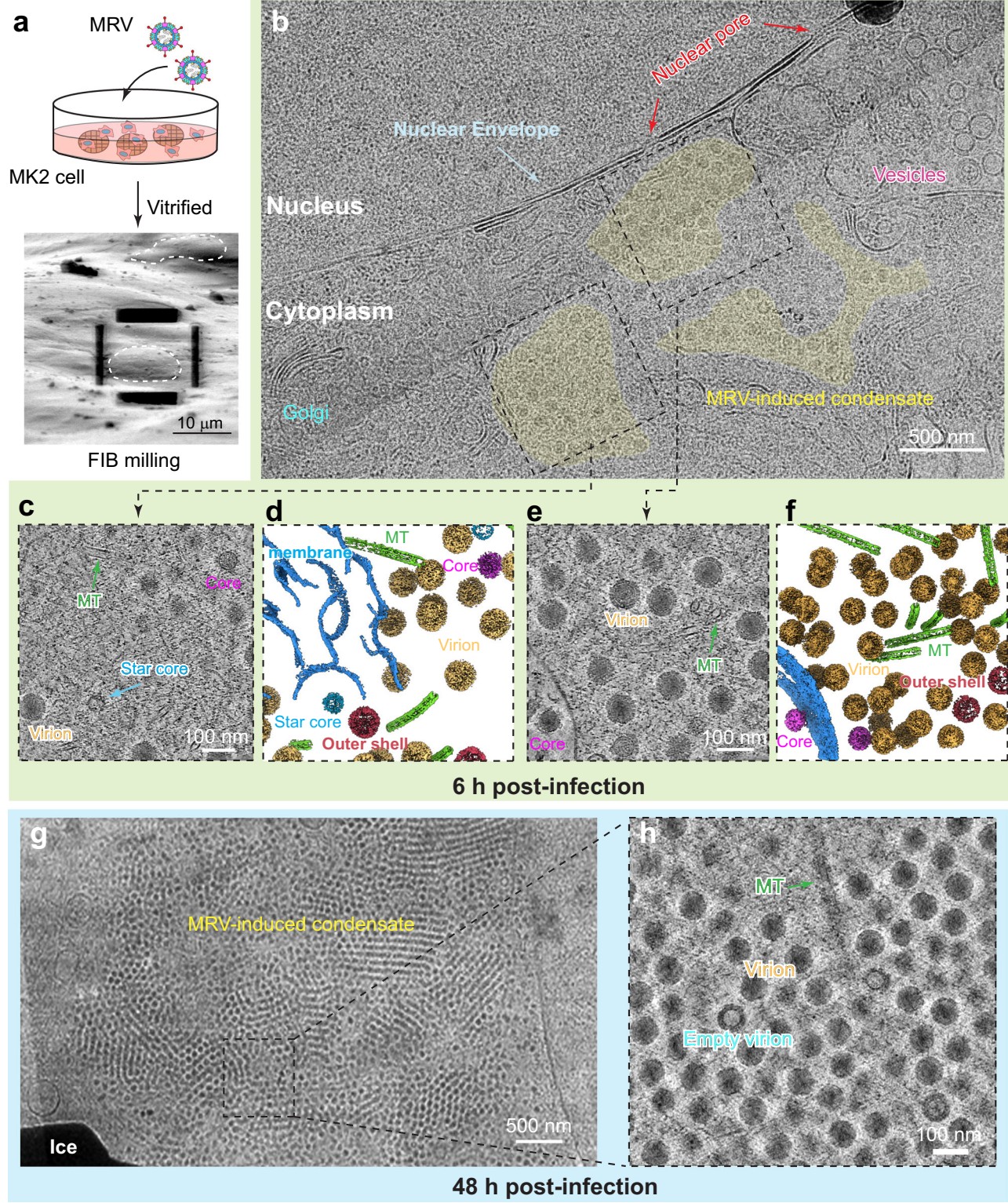

**Fig. 1 | MRV-induced cellular condensates in host cells visualized by cryoET.** **a** MRV-infected cells cultured on cryoEM grids for focused ion beam milling. Example cells in the scanning electron microscope (SEM) image are outlined by dashed lines. **b** Low magnification cryoEM image of the lamella from cells at 6 h post-infection. Dashed boxes indicate the areas for tilt series collection. Slice images (**c**, **e**) and corresponding 3D segmentation (**d**, **f**) from cryoET reconstructions, showing the condensates in host cell cytoplasm. **g** Low magnification cryoEM image of the lamella from cells at 48 h post-infection. Dashed boxes indicate the areas for tilt series collection. **h** A slice image of cryoET reconstruction from cells at 48 h post-infection. The experiments in (**a**, **b** and **g**) were repeated more than three times with similar results.

attaches first, followed by the stepwise attachment of μ1/σ3. The outer shell particle appears to be a byproduct of assembly, as it lacks any inner layer proteins or RNA genome. This notion is supported by our single-particle analyses of viral isolates (described below), which show a much higher percentage of outer shell particles in the late infection stage as compared to the early infection stage. These particles are likely assembled from the overexpressed and unused λ2, μ1, and σ3 in the late stage of the MRV replication cycle. Intriguingly, in our

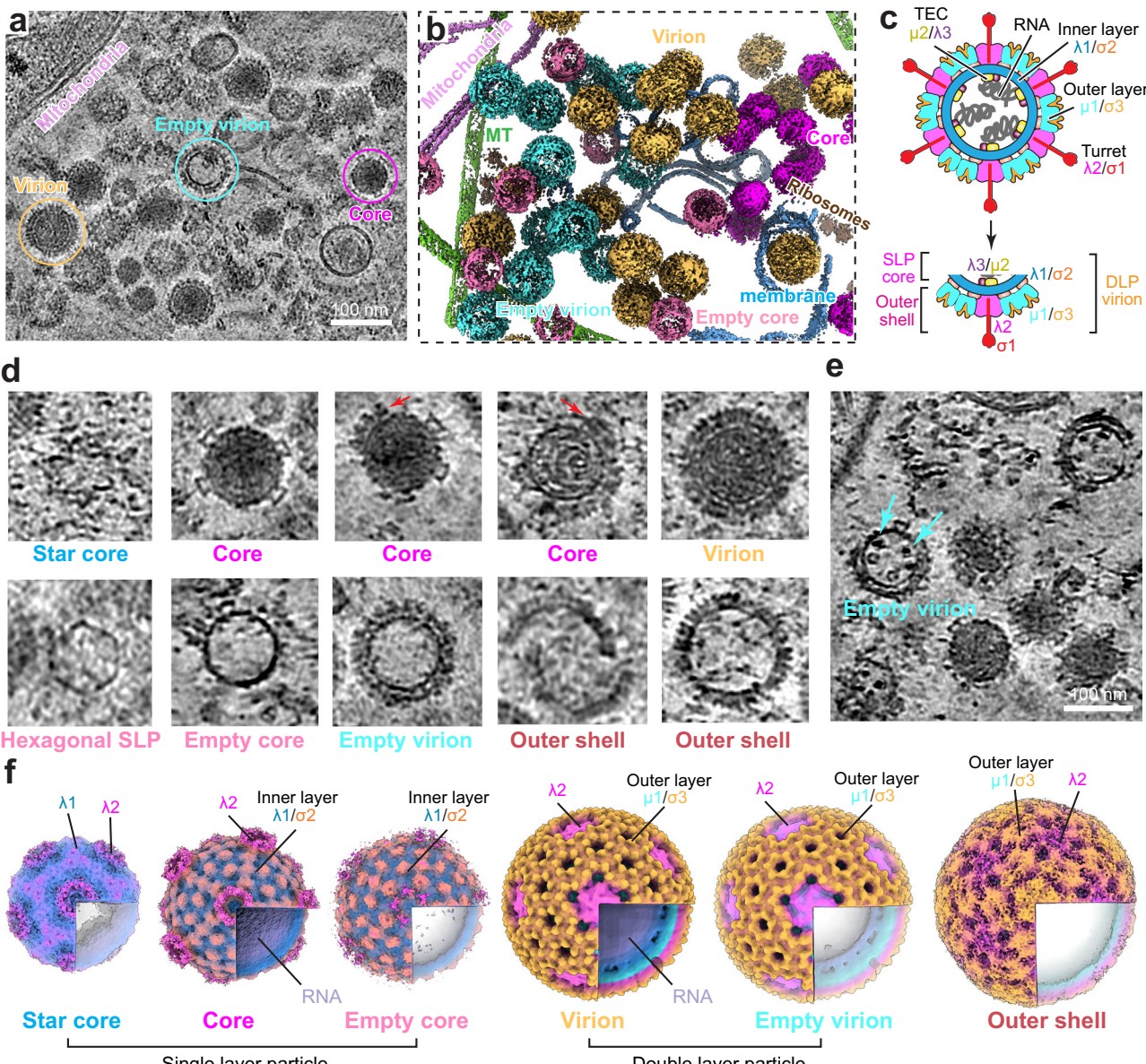

**Fig. 2 | MRV assembly intermediates and products captured by cellular cryoET.**
A slice image (**a**) and corresponding 3D segmentation (**b**) from cryoET reconstructions of cells at 6 h post-infection, showing virus assembly intermediates and products in the condensate, multiple tomograms show similar results. **c** Cartoon representation of the structure of MRV. Structural compositions of different viral particles (core, virion and outer shell) are indicated. SLP, single-layered particle; DLP, double-layered particle. **d** Representative assembly intermediate particles of MRV captured by cryoET. Red arrows point to the outer layer proteins. **e** A slice image of tomogram showing empty virion in the condensate. Cyan arrows point to the density of TEC inside the capsid, five similar particles were found in the tomogram. **f** Six intermediate states of MRV assembly captured by cellular tomography. CryoEM maps are colored by radius. Core and empty core are from data at 6 h post-infection. Star core, virion, empty virion, and outer shell are from data at 48 h post-infection.

tomograms, several genome-less particles, still clearly exhibiting densities inside the inner shell, were found (Fig. 2e, cyan arrows). We speculate that these densities belong to the TEC since the capsid can only contain the TEC and the genome inside. This suggests the TEC can be incorporated into the virus particle in a genome-independent manner. Overall, through cellular tomogram analysis, we identified six assembly intermediates and several transiently existing states, providing new and detailed snapshots of MRV stepwise assembly.

**Atomic structures of viral assembly intermediates and products**
However, the reconstructions of MRV assembly intermediates from these tomograms, with a resolution of up to 6.9 Å (Fig. S2), are still insufficient for detailed analysis of biochemical interactions at the atomic level. To obtain high-resolution structures of the assembly

intermediates and products described above, we performed cryoEM on cellular extracts containing viral particles, which were retained between two sucrose density layers after centrifugation of cellular extracts at two post-infection time points (6 h and 48 h). Asymmetric sub-particle reconstruction yielded near-atomic resolution (up to 3.0 Å) structures for five types of viral particles, all with visible side chain densities for atomic model building (Figs. 3, S3, and S4): from 6 h post-infection, core, virion, empty core, and empty virion were reconstructed (Figs. 3 and S3); and for 48 h post-infection, structures of virion, empty virion, and outer shell particle were obtained (Fig. S4). Analysis of total particle abundance from the two-time points shows that the proportion of single-layered particles decreases between the 6 and 48 h post-infection points (4.1% vs 1.3%), while the proportion of outer shell particles increases dramatically during that time (0.5% vs

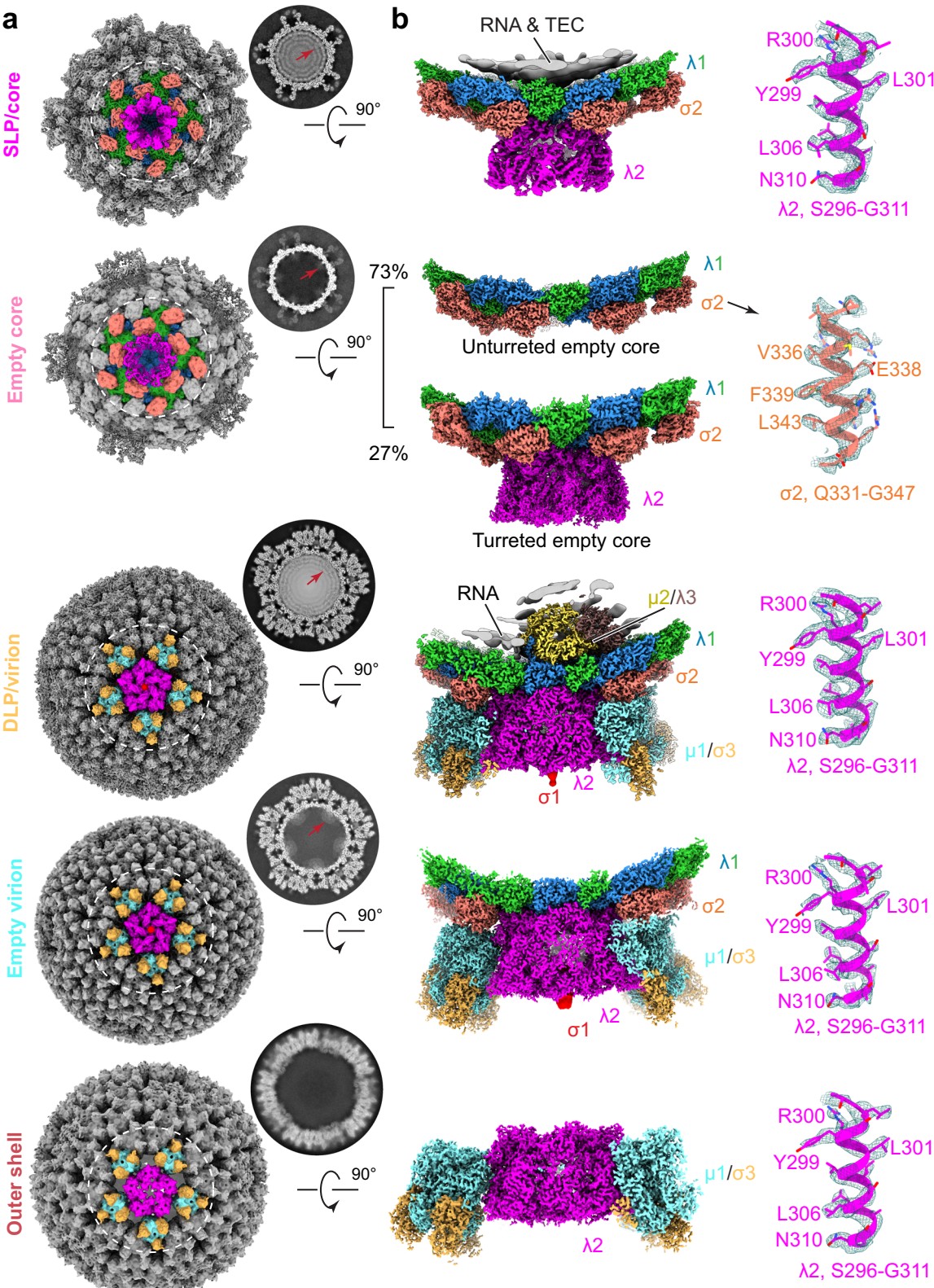

**Fig. 3 | High-resolution structures of MRV assembly intermediates resolved by single particle cryoEM analysis of viral isolates. a** CryoEM maps and the corresponding cross-sections of icosahedral reconstructions of five assembly intermediates. **b** Vertex reconstructions by sub-particle boxed out from the icosahedral map in (**a**). CryoEM densities of different proteins and RNA are segmented and colored differently as labeled in the figure. Superposition of density map (mesh) and atomic model, showing side-chain densities. Core, empty core, virion are from data of 6 h post-infection. Empty virion and outer shell are from data of 48 h post-infection.

22.5%) (Fig. S3, S4; Methods). The star core, which was observed by cellular cryoET (above), was not sorted out from extract of MRV-infected cells. This was likely due to its fragility, which would make it unable to survive the sucrose purification step, as well as its low abundance.

We subsequently compared the structures of genome-filled particles (core and virion) with those of genome-less particles (empty core and empty virion) and found that their structural features differ, both inside and outside the particle, at the fivefold axis. Inside the particle, reconstructions of virion and core particles clearly show that the layered RNA genome and densities of TEC (Fig. 3a, at red arrows) are underneath the inner shell, indicating that genome replication from ssRNA to dsRNA has been completed in the core. Within the empty virion and empty core, no RNA density beneath the inner shell at the fivefold axis was observed, but weak protein density, reminiscent of the TEC (Fig. 3a, at red arrow), appeared. However, an asymmetric reconstruction of the TEC was obtained after focused classification in virions only, not in the empty virion. This result suggests that the RNA genome is important for stable docking of TEC to the inner layer protein shell, though it is not required for TEC incorporation into the capsid, as also suggested by the tomograms. Due to the limited number of particles, we were unable to obtain an asymmetric reconstruction of the core with TEC resolved inside. Outside the particle, the pentamer of turret protein λ2 lies at the fivefold axis. All genome-filled

core particles have the turret attached at the fivefold axis. In empty (genome-less) core particles, however, the density of the turret is much weaker compared to that in core particles, as revealed by 3D reconstructions and cross-section images (Fig. 3a). This weak density is explained by focused classification of the empty core after sub-particle reconstruction, which shows that only 27% of particles have the turret attached (Fig. 3b). Thus, RNA genome may facilitate turret attachment. Therefore, the correlation of atomic structures and cellular tomograms enables us to understand not only the choreography but also the mechanism underlying the genome packaging and capsid assembly of MRV, as detailed below.

## Viral genome packaging and MRV virion assembly

The star core captured in our tomograms represents the early assembly state, which is crucial for understanding the genome packaging of MRV. While our structure of the star core resembles the star-shaped SLP (single-layered particle) of MRV captured in a previous cryoET study (Fig. S5)[32], which also shows a single-layered λ1 shell with a large invagination at the fivefold axis, it reveals an additional density above the invagination, outside the shell (Fig. 4c, indicated by the red arrow). This density fits well with the pentamer of λ2. The central channel of the pentamer, combined with the pore at the bottom of the invagination, collectively forms a tunnel that connects the interior and exterior of the star core (Fig. 4a). As ssRNA replicates into dsRNA,

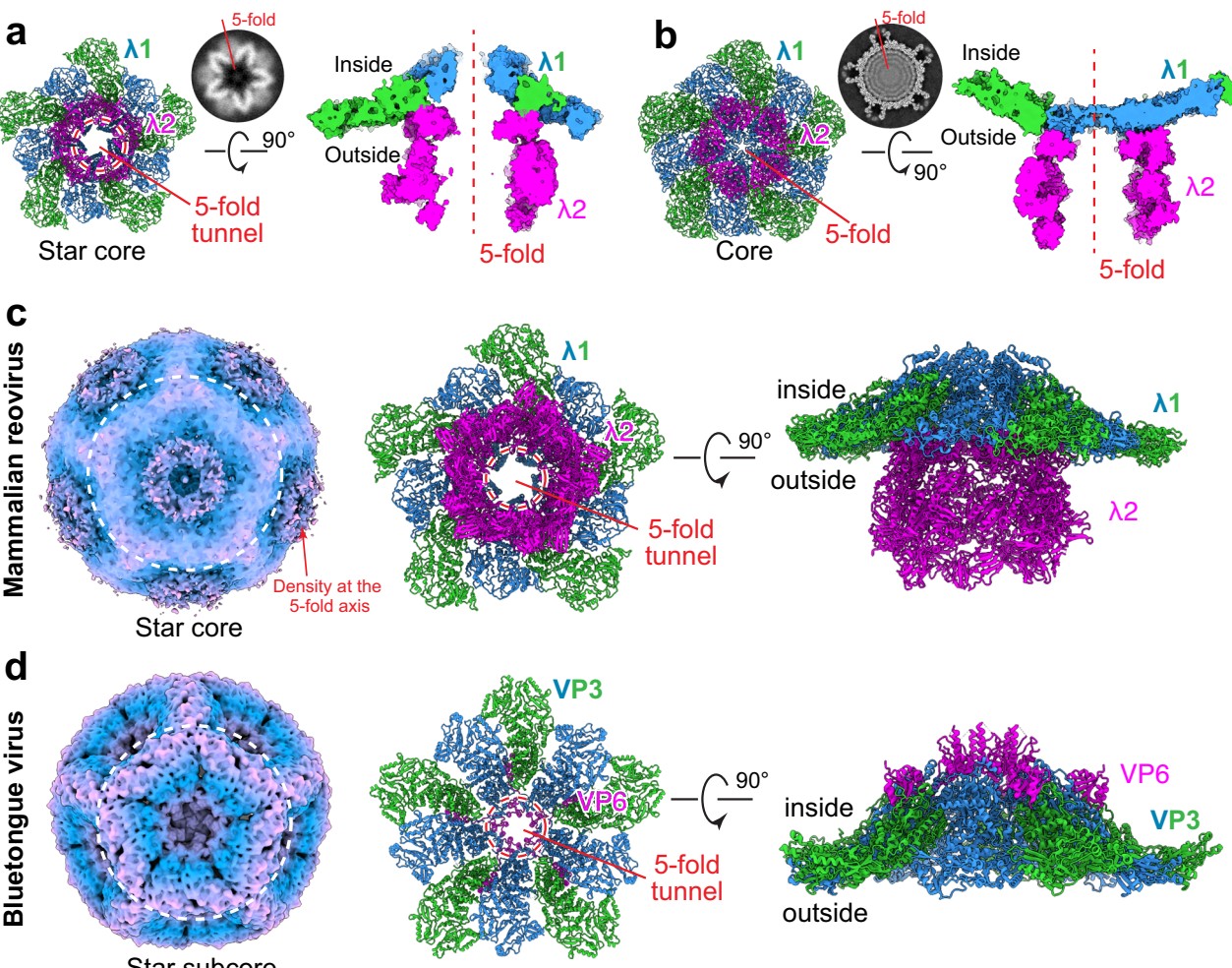

**Fig. 4 | Capsid propagation from star core to core during MRV assembly.** **a** Tunnel at the fivefold axis of the inner shell in MRV star core. The model is generated by docking structures of λ1 (PDB: 6XF7[32]) and λ2 (from core particle of this study) to the cryoEM map separately. **b** Atomic model of MRV core showing that the fivefold axis of the inner shell is closed. **c** CryoEM map of MRV star-shaped SLP and two orthogonal views of the fitted model at the fivefold axis. CryoEM map is colored by radius. **d** CryoEM map of BTV star-shaped SLP (EMD-43716) and two orthogonal views of the atomic model (PDB: 8W19) at the fivefold axis[15].

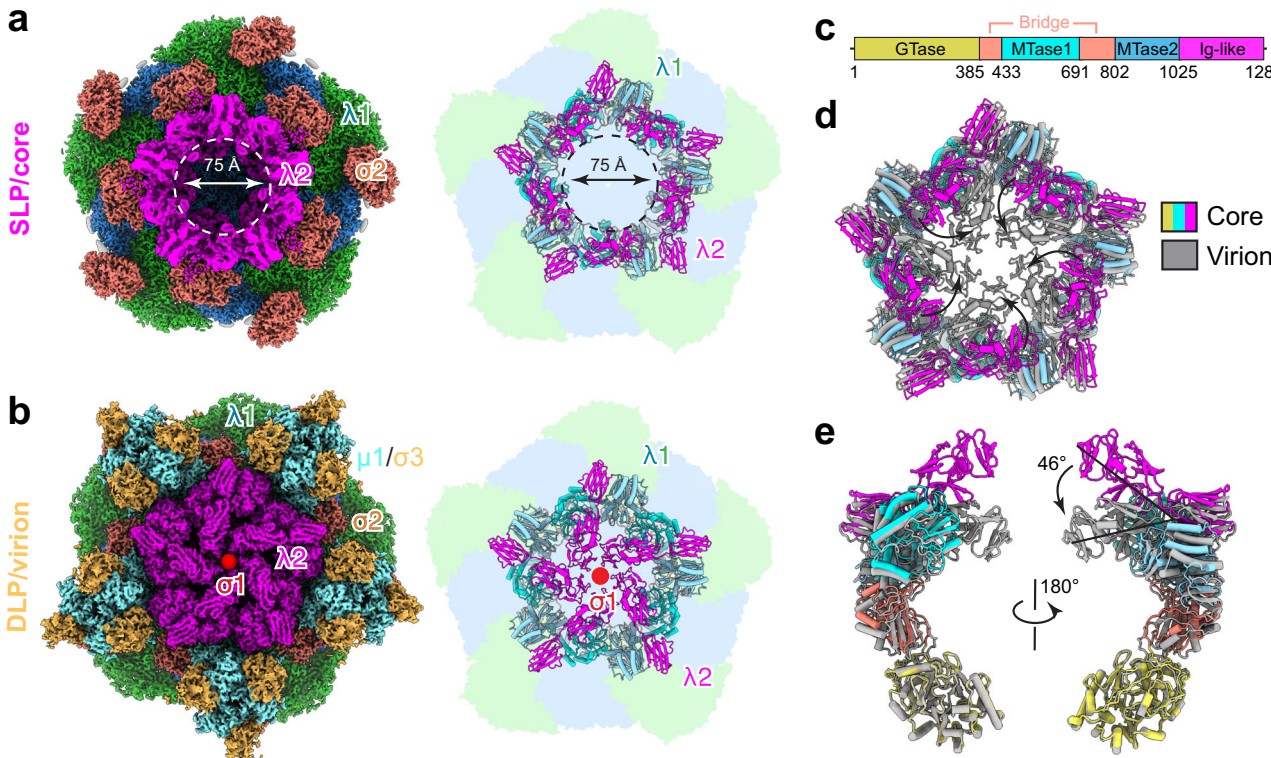

**Fig. 5 | Conformational change of the turret protein λ2 during assembly.**
**a** CryoEM map and the atomic model of MRV core from sub-particle reconstruction showing the turret is open with a central channel circled by dashed lines. The inner shell protein λ1 is represented as shadow and σ2 is omitted from the model for clarity. **b** CryoEM map and the atomic model of MRV virion from sub-particle reconstruction showing the turret is closed. The spike protein σ1, not built in the model, is indicated as a red dot at the fivefold axis. **c** Domain organization of λ2. Structure superposition of core and virion showing the conformational change of λ2 during virus assembly. The inner shell protein λ1 is omitted. λ2 pentamers are shown in (**d**) and one of the five subunits in orthogonal view is shown in (**e**). Movements of the Ig-like domain of λ2 from core to virion are indicated by arrows.

filling the core, the inner shell expands from a star shape into a spherical shape, pushing the walls of the shell to close off the tunnel at the fivefold axis (Fig. 4a,b; Movie S4).

The tunnel formed by turret protein λ2 and invaginated λ1 of MRV resembles the RNA binding tunnel formed by VP6 and VP3 in BTV (Fig. 4c,d)[15]. During core formation, VP3 also expands to seal off the RNA binding tunnel at the fivefold axis where VP6 is located, though on the inside of the shell rather than outside. Although λ2 shares no sequence or structural homology with VP6, the similar organization of the 5-fold tunnel implies that MRV may adopt a similar mechanism for RNA packaging as that of BTV. In this model, the unencapsidated ssRNA is threaded into the capsid through the fivefold tunnel during the early assembly stage. The RNA residing within the fivefold tunnel may facilitate the recruitment and attachment of the turret. Notably, cellular cryoET studies of rotavirus-infected cells also exhibited a star-shaped SLP with extra density outside the invaginated capsid at the fivefold axis (Fig. S5)[38], further supporting the idea that both turreted and unturreted members in the family of *Reoviridae* share a similar mechanism of genomic RNA packaging.

λ2 is recognized as the capping enzyme for newly synthesized mRNA in the core. Five λ2 subunits form a pentameric turret at each of the fivefold axes, serving as the mRNA exit channel[39]. A mechanistic interpretation was not possible for the previously observed conformational change in the turret between the core and virion[25] at low-resolution (30 Å). The atomic structures obtained here now allow us to clearly define these conformational changes. λ2 consists of a GTase domain, an MTase1 domain, an MTase2 domain, a bridge domain that connects the GTase and the methylases domains, and three Ig-like domains at the C-terminus (Fig. 5c). The λ2 pentameric turret changes conformation during virus assembly: in the core, the turret is open, with a central channel 75 Å in diameter (Fig. 5a); in the virion, it closes (Fig. 5b), sealing off the channel. In the open conformation, the Ig-like domains of λ2 straighten, forming the channel for mRNA capping and export. As the proteins μ1/σ3 coat the core to form the outer layer of the virion, the last two of the three Ig-like domains, acting as a rigid body, flip down 46° to close the channel, which is then sealed by σ1 attachment (Fig. 5d,e; Movie S5). Unlike the core observed in this study with an open turret, the crystal structure of the MRV core[31], which was obtained on samples generated from chymotrypsin-digested virions, shows the turret in the same closed conformation as that in the virion (Fig. S6). Studies have shown that assembled core particles in the condensates are active in mRNA transcription[30,37,40,41], but this activity is inhibited after outer layer proteins μ1/σ3 and σ1 are incorporated to form virion particles[41,42]. Our structural data align with these functional findings: in the core, the open conformation of the turret maximally facilitates mRNA export efficiency, while in the virion, the closed conformation of the turret suppresses transcription, leading to a virion in the quiescent state.

## Discussion
Our cryoET of MRV-infected cells establishes that viral factories are non-membrane-bound but membrane-containing cellular condensates introduced by, and subsequently used for, viral replication. The 3D tomograms at different time points post-infection reveal the spatiotemporal molecular sociology of these condensates within the cellular context: during infection, the sparsely distributed small condensates fuse into one large condensate that eventually dominates the cytoplasmic space; as the condensates form, they remodel

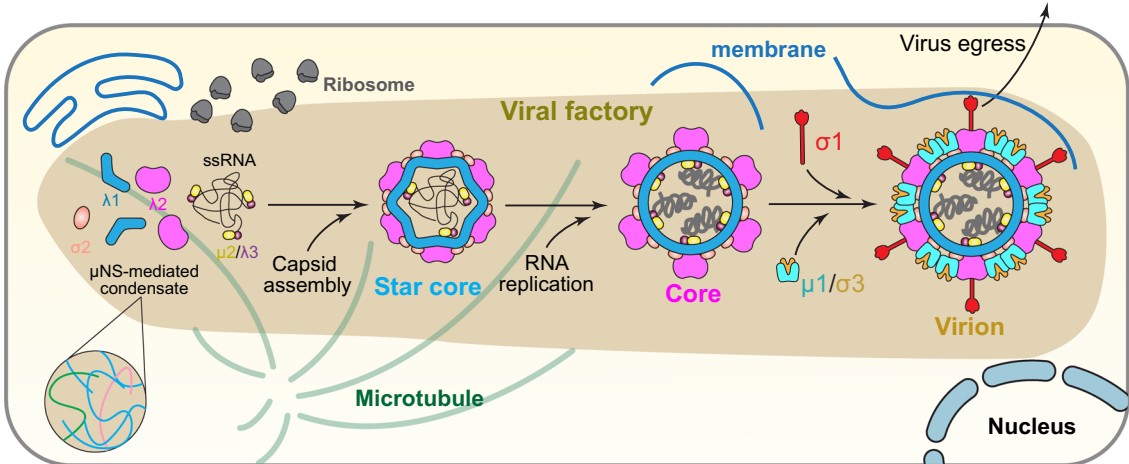

**Fig. 6 | Cartoon representation of MRV assembly and replication in cellular condensate.** Virus assembly and replication occur in µNS-mediated cellular condensates[62], which incorporate host machineries, viral proteins, and RNA. RNA replication expands the star core to a spherical core, followed by the assembly of outer layer proteins µ1/σ3 and the spike protein σ1. Note: Depictions are not drawn to scale; the elongated shape of the viral factory/condensate does not reflect its actual shape, which is pleomorphic. The zoomed-in inset at the bottom left corner depicts µNS interactions that are involved in forming MRV-induced condensates.

the existing cellular architecture, sorting and gathering host machineries within or around it, such as decorated and undecorated microtubules, lipid membranes, and ribosomes, for viral replication and egress (Fig. 6). The proportion of core particles to all particles decreases from the early to the late infection stage, while the proportion of mature virions, along with the byproduct outer shell particles, increases over that time. Viral intermediates and products, including core and virion, concentrate within the same condensate, gathering in a disorderly manner without a clear patterned spatial distribution.

By combining cryoET with cryoEM, we gained detailed insights into the in situ activities of MRV through a molecular sociological lens. We captured a series of MRV assembly intermediates at high resolution, along with several transient states, and revealed details of genome packaging and virus assembly from a genome-less star core, comprising a single layer, into a full, genome-filled, double-layered virion (Fig. 6). Inner capsid proteins interact with ssRNA genome segments to form the star core. During or shortly after inner-layer formation, ssRNA replicates into dsRNA[43], expanding the shape of the core from a star shape to a hexagonal shape and finally to a spherical shape. Simultaneously, genome incorporation facilitates turret attachment and is necessary for TEC stability; though the latter can be incorporated into particles in a genome-independent manner, it loosely docks underneath the inner shell. The coating of outer layer proteins µ1, σ3, and σ1 occurs continuously throughout capsid formation and completes the virion particles, which eventually egress from the cell. During the assembly process, the viral particle undergoes a series of conformational changes: a tunnel, formed by the inner shell and turret at the fivefold axis, closes at the inner shell as the star core becomes the core; the other end of the turret flips shut as the core becomes the virion. Our study presents the choreography underlying the genome packaging and capsid assembly of MRV.

A central question about condensates is how they are spatially organized and temporally regulated. This study demonstrates the power of integrating advanced techniques such as cryoET and cryoEM to study the MRV-induced cellular condensates at different time points, providing a "molecular sociology" perspective on the progression of MRV infection within cells. MRV infection halts host protein synthesis and begins viral protein synthesis by hijacking host machineries[44,45]. Despite all necessary materials—viral RNA, nonstructural proteins, inner layer proteins, and outer layer proteins—

being located within these condensates[30], processes such as transcription, translation, genome packaging, virus assembly, and maturation occur in a stepwise and coordinated manner. Our results show that the MRV-induced non-membrane-bound cellular condensates are highly regulated structures rather than sporadic aggregates. Viruses, with their limited number of genes, provide a controlled and simplified environment compared to the complexity of eukaryotic cells[46]. Importantly, this study showcases the internal architecture of virus-induced cellular condensates, which have relatively long lifetimes (h) with just a few distinct intermediates and exhibit high spatial resolution (Å) for atomic interpretation. Their simplicity and stability facilitate the tracking and dissection of these condensates to understand the organizational and functional principles, benefiting fields across virology, biophysics, and cell biology where molecular condensates or phase separation have been recognized to play important roles in various processes[47,48]. Thus, this investigation of MRV-induced cellular condensates highlights the potential of using viruses as a simple model to study condensate mechanisms and their relevance to both viral replication and broader cellular biology.

## Methods
### Virus culture
LLC-MK2 derivative cells (ATCC, CCL-7.1) were maintained in Eagle's Minimum Essential Medium (EMEM) supplemented with 10% Fetal Bovine Serum, 100 I.U. penicillin, and 100 µg/mL streptomycin. Cells were incubated at 37 °C and 5% $CO_2$ in air atmosphere. The cells in T175 flasks at 90% confluence were infected with MRV serotype Dearing 3 (ATCC, VR-824) at a multiplicity of infection (MOI) of 50, and harvested at 6 h post-infection, as an early stage infection sample. We used MOI of 10 to infect the cells and harvested the cells at 48 h post-infection, as a late-stage infection sample.

### Cell seeding on EM grids
Carbon-coated gold grids (Quantifoil R 2/2) were soaked in 70% ethanol and washed in fresh medium before cell seeding. Pre-treated grids were seeded with LLC-MK2 derivative cells in six-well plates and incubated for 24 h at 37 °C. Cells were then infected with MRV as described above. The grids were flash-frozen in liquid ethane using an FEI Vitrobot Mark IV (Thermo Fisher Scientific) with a blot force of 1 for 10 s at 6 h or 48 h post-infection, respectively.

## CryoFIB milling

Grids with vitrified cells were loaded into an Aquilos DualBeam (FIB/SEM) system (Thermo Fisher Scientific) at liquid nitrogen temperature. Before milling, the grid was sputter-coated with platinum, followed by a thick (~300 nm) layer of carbon-rich platinum applied by using the in-chamber gas injection system (GIS). Cells on the grid were identified in the low magnification SEM image, and those located in the center of squares were selected for FIB milling by gallium ion beam in a stepwise manner: preparation, trenching stress relief cuts, rough milling, medium milling, and manual polishing[15]. During the preparation step, each targeted cell was aligned to the eucentric position, and the shallowest possible milling angle, typically 8°–18°, was determined individually. All milling processes were conducted at an accelerating voltage of 30 kV. The beam currents used were as follows: 700 pA for milling stress relief cuts and rough milling, 100 pA for medium milling, and 10 pA for polishing (6 h post-infection); 1 nA for milling stress relief cuts and rough milling, 300 pA for medium milling, and 30 pA for polishing (48 h post-infection). Grids containing milled lamellae were transferred to liquid nitrogen dewars prior to cryoET tilt series collection.

## Acquisition and processing of cryoET tilt series

The good lamellae were loaded into a 300 kV Titan Krios electron microscope (Thermo Fisher Scientific). For the cells at 6 h post-infection, data were collected with a Titan Krios G4 equipped with a Selectris X imaging filter and Falcon 4 direct electron detector. Tilt series were collected at a nominal magnification of 42,000×, corresponding to a calibrated pixel size of 1.965 Å at specimen level. For the cells at 48 h post-infection, data were collected with a microscope equipped with a Gatan imaging filter (GIF) Quantum LS and a Gatan K3 Summit direct electron detector. Tilt series were collected at a nominal magnification of 33,000×, corresponding to a calibrated pixel size of 2.6 Å at specimen level. Tilt series were collected using a dose-symmetric tilt scheme starting from −10° between −60° and +40° with a tilt increment of 3° (or between −40° and +60° with a start at 10°) in serialEM[49]. Defocus was set between −3.5 μm and −5 μm and the total cumulative dosage for each tilt series was about 100 electrons/Å².

Frames in each movie were aligned in MotionCor2[50] and defocus value of individual image was determined by CTFFIND4[51]. After stack generation, tomograms were reconstructed in IMOD 4.11 software package[52] by using patch tracking and weighted back projection. Tomograms containing viruses with high image contrast and good alignment were selected. For visualization, tomograms were binned 4× and subjected to missing wedge compensation and denoising by using the deep learning-based software package IsoNet[53]. Tomograms were rendered in ChimeraX[54] by using the volume tracer. To improve the resolution, different assembly intermediates of MRV were picked separately in IMOD, and then imported into RELION 4.0[55] for sub-tomogram averaging (see Fig. S2 and Tables S1, S2).

For cryoET data of cells at 6 h post-infection, 20 high-quality tomograms were selected for further subtomogram averaging. Particles of different assembly intermediates were manually picked in IMOD. In total, 23, 13, 22, 125, 13, and 23 particles were picked for star core, empty core, core, virion, empty virion, and outer shell, respectively. The particle coordinates were subsequently imported into Relion 4.0[55] for averaging. For star core, core, and empty core, particles were extracted with box size of 192 pixels in bin2 (3.93 Å per pixel) and reconstructed with icosahedral symmetry to resolutions of 20 Å, 16 Å, and 20 Å, respectively. For virion, empty virion, and outer shell, particles were extracted with box size of 256 pixels bin2 (3.93 Å per pixel) and reconstructed with icosahedral symmetry to resolutions of 16 Å, 34 Å, and 38 Å, respectively. No further sub-particle reconstruction was performed for this data set.

For cryoET data of cells at 48 h post-infection, 25 tomograms were selected. 67, 29, 32, 1292, 432, and 43 particles were picked for star core, empty core, core, virion, empty virion and outer shell, and reconstructed with icosahedral symmetry to resolutions of 14 Å, 27 Å, 19 Å, 13 Å, 17 Å and 40 Å, respectively. Virion and empty virion were subsequentially subjected to sub-particle reconstruction to improve the resolutions. First, sub-particles from 12 vertices of the icosahedral reconstruction were extracted in bin1 (2.6 Å per pixel) with box size of 200 by using the angular and shift parameters from icosahedral refinement. 15,493 particles were obtained from 1292 virions and refined with C5 symmetry. After 3D classification without alignment to remove some bad particles, the data set was subjected to Tomo CTF refinement and Tomo frame alignment in Relion. This cycle of refinement (particle re-extraction, 3D refinement, 3D classification, Tomo CTF refinement, Tomo frame alignment) was repeated three times until no improvement was observed. A final map at a resolution of 6.9 Å was reconstructed from 11,968 vertex particles of virion. Particles of empty virion were processed in the same workflow and a map of 10.9 Å was obtained.

## Virus isolation from cellular extracts

Infected cells at the set time point post-infection were pelleted by low-speed centrifugation for virus isolation. The collected cells were lysed in 50 mM NaCl, 100 mM Tris-HCl, pH 8.8, 0.1% NP40 and protease inhibitor on ice for 15 min. The lysate was cleared by centrifugation at 2000 g for 10 min and the supernatant containing MRV virus particles was collected. The supernatant was loaded onto a double cushion with 4 mL 66% (w/w) sucrose at the bottom and 10 mL 50% (w/v) sucrose on the top (both in PBS, pH 7.4), followed by centrifugation using rotor SW28 at 100,000 g for 1 h at 4 °C. The band at the interface of the two cushions was collected and diluted ten times with PBS, pH 7.4, then centrifuged at 16,000 g for 10 min. The supernatant was collected and further centrifuged through a 1 mL sucrose cushion (30% w/v in PBS, pH 7.4) at 80,000 g for 1 h at 4 °C with rotor SW41 Ti. The resulting pellet was resuspended in 15 μL PBS, pH 7.4, and used for the following cryoEM analysis.

## Single-particle cryoEM sample preparation and data collection

For viral isolate at 6 h post-infection, 200 mesh Quantifoil 2/1 holy carbon grids were used. The blot time and drain time were set to 5 s and 2 s, respectively. For isolated virus particles from cells at 48 h post-infection, 3 μL resuspended sample was applied to a glow-discharged 300 mesh lacey Au grid with a continuous thin layer of carbon (Ted Pella). After incubating for 60 s, the grids were blotted with blotting force of 8 and blotting time of 10 s and then flash-frozen in liquid ethane using a Vitrobot Mark IV (Thermo Fisher Scientific). Single particle datasets were collected with a 300 kV Titan Krios electron microscope (Thermo Fisher Scientific) equipped with a Gatan K3 Summit direct electron detector and a Gatan imaging filter (GIF) Quantum LS (slit width set at 20 eV). Movies were recorded with SerialEM at a nominal magnification of 81,000× (pixel size 1.1 Å). Exposure time was set to 2 s, fractionated to 40 frames, resulting in a total dose of 50 electrons/Å². Defocus was set in the range of −1.8 to −2.6 μm. A total of 22,739 movies were collected for samples of 6 h post-infection and 12,101 movies for samples of 48 h post-infection.

## Single-particle cryoEM reconstruction

The workflow of single-particle analysis of the viral isolate from cells at 6 h post-infection is summarized in Fig. S3 and Table S3. Each movie was aligned in MotionCor2[50], and the defocus value was estimated in CTFFIND4[51]. Virus particles were auto picked by Topaz[56] and extracted in Relion[57,58]. To accelerate the computation, particles were binned 2× to a pixel size of 2.2 Å with a box size of 400 × 400 pixels. After 2D classification, a total of 138,287 good particles containing different assembly states were selected for further classification of 2D and 3D, leading to five assembly states: core (1457 particles), empty core (3355 particles), virion (105,702 particles), empty virion (6587 particles) and

outer shell (531 particles). The strategy of sub-particle reconstruction was subsequently used to improve the resolution of core, empty core, virion, and empty virion. Briefly, sub-particles of 12 vertices were extracted in bin1 at pixel size of 1.1 Å and box size of 384 × 384 pixels by using the angular and shift parameters from icosahedral reconstructions. The extracted sub-particles were then subjected to further classification and refinement.

For core, a map of 3.3 Å with C5 symmetry was obtained from 10,857 particles. In each MRV particle, σ2 binds to the inner shell protein λ1 in three different configurations: $σ2_1$ and $σ2_2$ located inside the asymmetric unit, while $σ2_3$ located at the icosahedral twofold axis, shared by two asymmetric units. The density of $σ2_3$ was not resolved due to symmetry mismatch. To resolve this problem, a spherical mask at $σ2_3$ was applied for further 3D classification without alignment, resulting in two conformations of σ2 positioned in opposite direction, 48.7% and 51.3% particles classified to each of the two conformations. In conclusion, $σ2_1$ and $σ2_2$ adopt a uniform configuration in the asymmetric unit, and $σ2_3$ occupies two equally distributed opposite directions, shared by two asymmetric units, leading to a total of 150 σ2 copies in one MRV particle ($σ2_1 × 60 + σ2_2 × 60 + σ2_3 × 60/2$).

For empty core, the density of the turret is relatively weak compared to that of core, indicating low occupancy of turret protein λ2 in the empty core in the absence of genomic RNA. By 3D classification with a mask at the turret, two different conformations of empty core were obtained: unturreted empty core without λ2 and turreted empty core with λ2. After further refinement, unturreted empty core was reconstructed to 3.6 Å with C5 symmetry from 26,781 particles and the turreted empty core was reconstructed to 3.8 Å with C5 symmetry from 8093 particles.

To resolve the density of TEC inside the virion, vertex particles after C5 refinement were expanded five times, followed by skip-align 3D classification with spherical mask at the TEC region. One set of particles containing 97,187 particles representing one of the five directions of TEC was selected. After final C1 refinement, the map of virion with clear TEC density was reconstructed to 3.0 Å. Sub-particle reconstruction of empty virion was obtained with C5 symmetry at a resolution of 3.9 Å from 60,524 particles. Smeared density of TEC in the empty virion was also observed. However, a similar strategy of C1 reconstruction as that used for virion failed to resolve the TEC, indicating that genomic RNA is required for the stable binding of TEC onto the inner capsid shell. For the outer shell, due to the particle heterogeneity and small particle number, sub-particle reconstruction failed to improve the cryoEM density of outer shell.

The cryoEM dataset of viral isolate from cells at 48 h post-infection was processed similarly as described above for the isolate from cells at 6 h post-infection, except the particles were picked manually and is summarized in Fig. S4 and Table S4. Briefly, 12,101 micrographs were motion-corrected for particle picking. 39,141 particles were picked, and 31,913 particles were selected after 2D classification. Additionally, 415 particles of MRV SLP core were picked. Due to the low particle number and poor quality, the processing did not yield useful reconstructions. In the following 3D classification, three different assembly states were separated: 19,870 particles in virion, 3685 particles in empty virion and 6834 particles in outer shell. These particles were subsequently expanded 12 times in the vertex sub-particle extraction, followed by 3D classification and refinement, resulting in final maps of virion, empty virion and outer shell with C5 symmetry at resolutions of 3.0 Å, 3.2 Å, and 3.7 Å, respectively.

### Atomic modeling, model refinement, and structure visualization
Model building of virion, outer shell, and core was started by using the cryoEM structure of ISVP as an initial model[24]. For virion, models of the inner shell protein λ1, σ2, transcription enzyme complex μ2/λ3, turret protein λ2, and outer layer proteins μ1/σ3 were docked into the cryoEM structure of virion from early infection (6 h post-infection) stage[31]. The model was manually adjusted in Coot[59] according to the density and refined in ISODLE[60] and Phenix[61]. The refined model of MRV virion was used for model building of the core and outer shell. For turret protein λ2 in the core, structures of GTase, bridge, MTase1, MTase2, and Ig-like domain from virion were sequentially fitted into the cryoEM map of core and adjusted in Coot. The map of the core was Gaussian filtered in ChimeraX with a standard deviation of 1.8 to fit the Ig-like domain. All models were refined in ISOLDE and Phenix and validated by the wwPDB validation server. Visualization of the atomic model, including figures and movies, was accomplished in UCSF ChimeraX[54].

### Reporting summary
Further information on research design is available in the Nature Portfolio Reporting Summary linked to this article.

## Data availability
The cryoEM density maps were deposited in the Electron Microscopy Data Bank (EMDB) with accession code: core (EMD-46053), unturreted empty core (EMD-47321), turreted empty core (EMD-47322), virion (EMD-46054) from dataset at 6 h post-infection; empty virion (EMD-47320) and outer shell (EMD-46049) from 48 h post-infection; core (EMD-47313) and empty core (EMD-47314) from subtomogram averaging of the dataset at 6 h post-infection; star-core (EMD-47315), virion (EMD-47316, EMD-47317), empty virion (EMD-47318), outer shell (EMD-47319) from subtomogram averaging of the dataset at 48 h post-infection. The corresponding models were deposited in the Protein Data Bank (PDB) with core (9CYX), virion (9CYY), and outer shell (9CYT). This paper does not report the original code.

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

## Acknowledgements

We thank Titania Nguyen, Miriam Lepiz, and Nicolas Coral for editing the manuscript. The lamellae sample and cryoET data of MRV infection at 6 h post-infection were collected at the Midwest Center for CryoET (MCCET) and the CryoEM Research Center located in the Department of Biochemistry at the University of Wisconsin-Madison, supported by the NIH Common Fund Transformative High-Resolution Cryo-Electron Microscopy program (U24 GM139168 to Elizabeth Wright). This project is supported by grants from the US NIH (AI094386 to Z.H.Z.). We acknowledge use of resources in the Electron Imaging Center for Nanomachines supported by UCLA and grants from the NIH (1S10OD018111) and the National Science Foundation (DBI-1338135 and DMR-1548924) and the cryogenic focused ion beam (cryoFIB) instrument in the Gonen lab supported by NIH (P41GM136508), the Department of Defense (HDTRA1-21-1-0004) and Howard Hughes Medical Institute.

## Author contributions

Z.H.Z. conceived the project and supervised research; X.L. and X.X. prepared samples, recorded cryoEM and cryoET images and processed the data; M.M. and T.G. generated lamellae of infected cells at late stage grown on grids by X.L.; X.L., X.X., and Z.H.Z. interpreted results and wrote the manuscript; all authors reviewed and approved the submitted paper.

## Competing interests

The authors declare no competing interests.
