## [Transparent Peer Review file · Nature Communications]

Molecular Sociology of Virus-induced Cellular Condensates Supporting Reovirus Assembly and Replication

Corresponding Author: Professor Z. Hong Zhou

Version 0:

Reviewer comments:

Reviewer #1

(Remarks to the Author)

This manuscript describes the atomistic structures of intracellular assembly intermediates of mammalian reovirus (MRV) through a combination of cryo-EM subtomographic averaging of cellular particles and single-particle analysis of biochemically enriched intracellular virus particles. The manuscript presents many significant findings, both minor and major, with the most noteworthy being the provision of atomic models of the active open conformation of turret $\lambda 2$ protein located on the surface of single-layered assembly intermediates in the star core and core.

The methods employed in this manuscript are based on a recently published paper by the same group (Xia et al., Cell 2024), which is well-written and thoroughly validated. Determining in situ intracellular functional structures of viruses is currently a highlighted topic in the field of virus research, and thus, their methods and findings have significant impacts on the field. While the structure of assembly intermediates has been studied for a long time, the level of detail achieved in this manuscript has not been previously attainable (e.g., a relatively recent paper by Sutton et al., Nat Comm 2020). Elucidating the structural functions of the turret $\lambda 2$ protein, the 5-fold gate functions for RNA transcripts, and the hypothesized intraparticle genome synthesis mechanism are central to understanding dsRNA virus transcription and replication. The atomistic details, along with the open and closed conformations, corroborate the hypothesis connected with reovirus core functions. Due to these critical biological findings and methodological advances, I am inclined to be positive about their results and conclusions. However, I have some minor comments.

Minor Comments

1. Lines 121-122: It is not clear why these particles are considered for gene therapy. Are there any concrete ideas or studies supporting this?
2. Lines 149-152: Why is the TEC not clearly visible in the SLP/core, unlike in the mature DLP/virion? Is there a reasonable explanation for this?
3. Lines 162-164: Are there any structural differences between filled and empty core particles that might explain the difficulty in recruiting turret proteins to the core? It seems likely that RNA is not directly involved in the interaction between the turret proteins and the core.
4. Lines 182-184: Is the tunnel of the $\lambda 2$ protein highly positively charged? Considering the highly positively charged tunnel structure of the BTV VP6, it is crucial to display the positively charged residues in the tunnel of the MRV turret $\lambda 2$ protein. There is no explanation of what the α -helix structures in Fig. 3b represent. This should be mentioned in the figure legend. However, my guess is that this α -helix forms part of the turret protein's tunnel. These positively charged residues are likely central to the RNA-recruiting and ejecting functions.
5. Lines 196-198: The SLP/core and the open turret conformation are critical for synthesizing viral ssRNA transcripts, while the SLP/core is a transient assembly intermediate that eventually forms a transcription-inactive DLP/virion. Some particles would need to synthesize transcripts seamlessly. Are there any SLP/core particles dedicated to synthesizing viral transcripts but unlikely to assemble further, such as uncoated virus particles after entry? In your cryo-EM classifications of SLP/core particles, are there any structural variations in the SLP/core? In Fig. S3, it appears that two conformations of the SLP/core $\sigma 2$ proteins are shown (48.7% and 51.3%). Is there any relevance to this topic?

Reviewer #2

(Remarks to the Author)

The manuscript by Liu et al. investigates the architecture and dynamics of several mammalian reovirus (MRV) assembly intermediates at both early and late stages post-infection using a combination of cryo-electron tomography (cryo-ET) on cryo-focused ion beam (cryo-FIB) SEM lamellae and cryo-electron microscopy (cryoEM). This study elucidates the molecular interactions between these intermediates and their interplay with cellular components, as well as the temporal sequence and spatial rearrangement among them. The manuscript presents a significant piece of high-quality structural work with a well-described experimental approach that is both sound and robust. The data have been rigorously analyzed, and the paper is well-written. The results contribute significantly not only to the field of MRV research but also to the broader understanding of reoviridae assembly. The manuscript merits publication in its current form with a few revisions:

1. Page 6: The presentation of different single-layered particles in the first paragraph (line 102: star core, core, and empty core) and the second paragraph (line 109: star core, hexagonal core, and spherical core (referred to as core hereafter)) is confusing. This terminology should be clarified and homogenized. Additionally, the hexagonal core is introduced at this point in the manuscript but is not mentioned afterward. It is unclear how it "disappears," and this should be explained.
2. Page 6, Line 111: The sentence "Both the star core and hexagonal core contain internal densities, though only the spherical ones are fully occupied" requires clarification. The specific internal densities being referred to and what is meant by "fully occupied" should be explained in more detail.
3. Page 20, Lines 447-456: Data deposition codes are not specified. Additionally, all maps and PDBs described in the manuscript must be deposited in the database. For example:
 - o Figure 1 presents six intermediate states of MRV assembly captured by cellular tomography, but only the star core (EMD-XXXXX) and virion (EMD-XXXXX) from subtomogram averaging of the dataset at 48 hours post-infection are deposited.
 - o According to the manuscript, cryoEM data from 6 hours post-infection reveal the reconstruction of core, virion, empty core, and empty virion (Figures 3 and S3), and at 48 hours post-infection, structures of the virion, empty virion, and outer shell particle were obtained (Figure S4). However, only the core (EMD-XXXXX), unturreted empty core (EMD-XXXXX), turreted empty core (EMD-XXXXX), and virion (EMD-XXXXX) from the dataset at 6 hours post-infection; and empty virion (EMD-450XXXXX) and outer shell (EMD-XXXXX) from the 48-hour post-infection dataset have been deposited.
4. Figure S2: Equivalent particles at different time points appear significantly different (e.g., 6 hpi virion vs. 48 hpi virion). It is unclear whether these differences are genuine and should be addressed in the manuscript, or if they are due to differences in representation (e.g., the color scale used in each 3D rendering). This should be clarified.

Reviewer #3

(Remarks to the Author)

The paper presents a thorough structural examination of mammalian reovirus assembly intermediates visualized in cells isolated from virus-infected cells. These studies add more exciting data to the existing recently published literature, where similar studies of virus replicative intermediates characterized by FIB milling, cryo-ET, and SRP have been done with other members of the Reoviridae family, including BTV, rotaviruses, and reoviruses. I enjoyed reading the section on RNA incorporation at the 5-fold axes. I felt these assembly details should be the focus of the paper rather than the cell-virus interactions only touched on in the first section. However, my main criticism concerns that the authors make a point of studying the condensate without showing any data that reovirus replicative organelles are formed through LLPS, or present a condensate. The title describes "molecular sociology" to describe cell-virus interactions. This should be more clearly stated in the title. Most of the content is on MRV assembly stages and packaging ("choreography" as they neatly put it), which I think would make for a more representative title. Overall, the manuscript reads as another nice piece of work that addresses the visualization of viral intermediates in cells using cryo-ET, and it would be misleading to state that it sheds light on how the viral condensates are formed, or how they support viral replication, nor they reveal such dynamics of the interactions within the condensates, or with other cellular organelles ('sociology').

Broader points:

-Terminology and wording surrounding condensates were a bit loose throughout the text, including evidence. They should give their definition of a condensate-membrane-less inclusion. Is LLPS a requirement, or are other processes that form inclusions also capable of making condensates? Size requirement? Liquid-like only, what about changes in properties over time/as viral arrays form?

Evidence of MRV liquid-like properties (still not definitively LLPS) is not mentioned. PMC8406312 mentions LLPS of the MRV muNS protein, although more work needs to be done to address this question, as MRV replicative organelles have not been formally described as 'condensates' in literature yet.

-Important literature on MRV not touched on, including related evidence of LLPS. Lipid droplets or cellular membranes incorporated in condensates are not expanded on.

For example:

membrane plays important role in the replication factory of MRV: Tenorio et al show muNS and sigmaNS remodel ER membrane for interaction with inclusion bodies

doi: 10.1128/mBio.01253-18; or 10.1371/journal.ppat.1010641 These important papers are not discussed and may go against their condensate definition.

Figs 1 & S1- viral factories are not spherical, despite sphericity being used as an argument for LLPS in the introduction. This should be discussed, and some correlative imaging of condensates should be added to strengthen their argument that these

are indeed condensates.

Figure 6 - word coloring hard to read (blue of mu1 and yellow of viral factory). Condensate elongated shape may be misleading. What is the zoomed-in section on the bottom left referring to?

By line:

2- not all of the replication cycle necessarily takes place in factories.

-not all viral factories are condensates, e.g. membrane-bound replication organelles.

-Narrowing down to particular taxa or viral genome type (e.g. dsRNA) may be needed.

8- should change "dsRNA virus" to MRV specifically, same rules/mechanisms don't apply for all.

10: condensate not proven to "enable" replication and assembly, more studies showing impairment with disrupted MRV condensate are needed.

28: The definition of condensate needs improving. Condensates are not phase separation (= a process), nor a granule. These terms are not interchangeable

32: cellular condensates not indispensable for all viral lifecycles. e.g. rotavirus can still replicate, just impaired

36: The viral condensates referenced should be explicitly stated as LLPS properties don't apply to all. Also, growth/fusion shows liquid-like but not definitively LLPS.

66: temporal feels like a bit of a stretch given they rely entirely in this section on 2 images taken at 6h and 48h. No dynamics can be understood, lots of things can happen between these points that are lost.

70: 'Clearly show that MRV VFs are non-membrane-bound condensates with an ambiguous boundary' – does this mean it's not as clear if the boundary is ambiguous? Why is the shape not spherical if it's supposed to be a spherical droplet?

73: Same principle, due to lack of temporal information there is no evidence of "merging" to form large condensate from smaller. Ostwald Ripening is also possible. Either way, these data only show static snapshots of small regions of interest in frozen cells – there's no evidence of 'merging'.

84. a statement that dynein 'transports nascent virions' – the reference provided refers to unrelated viruses (Rabies for Ref 28), and without additional live-cell imaging studies with decorated dyneins or their cargos, it's impossible to tell what these densities are at this resolution. I would remove this speculative statement, or tone it down.

89. 'ribosomes in the vicinity of condensates' – I would discuss this result in light of this work published by John Parker's group: <https://doi.org/10.1128/mbio.01463>

92. I don't find any "sociology" between virus and cell in this section, just colocalization. No basis for the "hijacking" is also given despite being mentioned here - another study used nocodazole to show microtubule association and importance previously, and this is given in reference 27 published over 20 years ago. No data shown to support that nearby ribosomes are particularly responsible for viral protein synthesis, or the role of cell membrane etc. in condensates.

144. Please drop the term 'viral isolate' as this term has a different meaning in virology, and it can be confusing.

164: The authors should discuss how the RNA genome facilitates turret attachment in their model. Does this occur directly, or by changing shell shape/stability to enable attachment?

205: The statement that assembled core particles in the condensate are active in mRNA transcription has two references (32, 33) that do not show anything about transcription within condensates. Both ref. 32 & 33 show that cores are transcriptionally active in vitro, and when the reovirus muNS protein is added to cores and sticks to them (Ref.33). Unless the authors have their own data to demonstrate that they can visualize transcription in situ inside viral factories, they should remove this statement.

250: 'Cellular condensates are highly regulated structures rather than chaotic accumulations' – this is not what the paper shows, as many previous studies on condensates have already demonstrated that condensates are not aggregates. I would also use the term 'aggregate' instead of 'chaotic accumulation'.

Version 1:

Reviewer comments:

Reviewer #1

(Remarks to the Author)

The author addressed to all of my comments. I do not have anything more to add.

Reviewer #2

(Remarks to the Author)

The Revised manuscript address all my previous comments thoroughly and, in general, the revisions have significantly improved the clarity and quality of the manuscript. I have no further comments or suggestions and I recommend the manuscript for publication in its present form.

Reviewer #3

(Remarks to the Author)

Congratulations to the authors who have satisfactorily revised their manuscript, and it should be published in the revised version.

Responses to Reviewers (NCOMMS-24-45998-T)

Summary: We thank all three reviewers for their thoughtful comments, helpful suggestions and for supporting our work. As you will see from our itemized responses below, we have fully addressed the reviewers' questions. To facilitate your navigating our responses, we have copied the totality of the original reviewer comments in **black**, and written our responses in **blue**. The line numbers of changed text in the revised manuscript are indicated at the end of each answer (Ans). For your convenience of comparison with the original text, we also included a manuscript file with tracked changes highlighted in **red**.

Reviewer #1 (Remarks to the Author):

This manuscript describes the atomistic structures of intracellular assembly intermediates of mammalian reovirus (MRV) through a combination of cryo-EM subtomographic averaging of cellular particles and single-particle analysis of biochemically enriched intracellular virus particles. The manuscript presents many significant findings, both minor and major, with the most noteworthy being the provision of atomic models of the active open conformation of turret $\lambda 2$ protein located on the surface of single-layered assembly intermediates in the star core and core.

The methods employed in this manuscript are based on a recently published paper by the same group (Xia et al., Cell 2024), which is well-written and thoroughly validated. Determining in situ intracellular functional structures of viruses is currently a highlighted topic in the field of virus research, and thus, their methods and findings have significant impacts on the field. While the structure of assembly intermediates has been studied for a long time, the level of detail achieved in this manuscript has not been previously attainable (e.g., a relatively recent paper by Sutton et al., Nat Comm 2020). Elucidating the structural functions of the turret $\lambda 2$ protein, the 5-fold gate functions for RNA transcripts, and the hypothesized intraparticle genome synthesis mechanism are central to understanding dsRNA virus transcription and replication. The atomistic details, along with the open and closed conformations, corroborate the hypothesis connected with reovirus core functions. Due to these critical biological findings and methodological advances, I am inclined to be positive about their results and conclusions. However, I have some minor comments.

Ans: Thank you for your generous support of our work! Your comments are greatly appreciated and fully addressed below.

Minor Comments

1. Lines 121-122: It is not clear why these particles are considered for gene therapy. Are there any concrete ideas or studies supporting this?

Ans: We have deleted the speculative statement in the revised manuscript. See line 138.

2. Lines 149-152: Why is the TEC not clearly visible in the SLP/core, unlike in the mature DLP/virion? Is there a reasonable explanation for this?

Ans: We believe the **smear** TEC density in the SLP/core is mostly likely due to limited particle number (1,457 particles), which is far fewer than the DLP/virion (105,702 particles). To clarify this, we added one sentence in the revised manuscript: "Due to the limited number of particles, we were unable to obtain an asymmetric reconstruction of the core with TEC resolved inside." See lines 174-175.

3. Lines 162-164: Are there any structural differences between filled and empty core particles that might explain the difficulty in recruiting turret proteins to the core? It seems likely that RNA is not directly involved in the interaction between the turret proteins and the core.

Ans: There are subtle changes near the region of the 5-fold axis between filled and empty core particles. The filled core shifts approximately 3 Å outward compared to the empty core in this region (Rebuttal-Figure 1). In the two structures, the region for turret attachment remains unchanged. Therefore, we believe it is not the reason preventing turret recruitment.

The similar architectures of MRV and BTV star core at the 5-fold axis suggest that they may use a similar genome packaging mechanism. In this model, the unencapsidated ssRNA is threaded into the capsid through the 5-fold tunnel during the early assembly stage. The RNA residing within the 5-fold tunnel may facilitate the recruitment and attachment of the turret. We added the hypothesis in the revised manuscript. See lines 201-203.

4. Lines 182-184: Is the tunnel of the $\lambda 2$ protein highly positively charged? Considering the highly positively charged tunnel structure of the BTV VP6, it is crucial to display the positively charged residues in the tunnel of the MRV turret $\lambda 2$ protein. There is no explanation of what the α -helix structures in Fig. 3b represent. This should be mentioned in the figure legend. However, my guess is that this α -helix forms part of the turret protein's tunnel. These positively charged residues are likely central to the RNA-recruiting and ejecting functions.

Rebuttal-Figure 1: Structural comparison between the filled and empty cores showing subtle changes at the 5-fold axis.

Ans: The tunnel of the $\lambda 2$ protein exhibits scattered positively charged residues, though it is not as much as in the VP6 tunnel in BTV (see Rebuttal Fig. 2a). The RNA binding in this tunnel of MRV still needs further investigation.

There seems to be some confusion regarding the α -helix shown in Fig. 3b. The helix colored in magenta is located on the outer surface of the tunnel shown in the Rebuttal Figure 2b. The model and map superpositions are used to show the high quality of our reconstructions. The figure legend for Fig. 3b has been modified for clarity.

Rebuttal-Figure 2: a, Electrostatic surfaces representation of the $\lambda 2$ tunnel, with positive, neutral and negative Coulomb potentials indicated in blue, white, and red, respectively. b, Location of the helix shown in Fig. 3b. The helix is colored in magenta.

5. Lines 196-198: The SLP/core and the open turret conformation are critical for synthesizing viral ssRNA transcripts, while the SLP/core is a transient assembly intermediate that eventually forms a transcription-inactive DLP/virion. Some particles would need to synthesize transcripts seamlessly.

(Question a) Are there any SLP/core particles dedicated to synthesizing viral transcripts but unlikely to assemble further, such as uncoated virus particles after entry?

(Question b) In your cryo-EM classifications of SLP/core particles, are there any structural variations in the SLP/core?

(Question c) In Fig. S3, it appears that two conformations of the SLP/core $\sigma 2$ proteins are shown (48.7% and 51.3%). Is there any relevance to this topic?

Ans: For **Question a**, we don't have data to address it. In the literature, the assembly from transcription-active core to transcription-inactive virion seems to be regulated within cells. A paper published by Max Nibert's group (Broering et al., J. Virol., 2000) reported that non-structural protein μ NS bound to cores and μ NS-core complexes remained to be transcription active. Furthermore, in vitro competition assays showed that mixing μ NS with cores greatly reduced the formation of re-coated cores by the binding of outer-capsid proteins $\mu 1$ and $\sigma 3$. These results suggest that by binding to cores in the infected cell, μ NS may block or delay outer-capsid assembly and allow continued transcription by these particles¹. Another paper published by Maya Shmulevitz's group (Kniert et al., PLoS Pathog., 2022) reported that lipid droplets participate in the delayed assembly of outer-capsid proteins².

The answer to **Question b** is NO. In our cryo-EM classifications of SLP/core particles, no structural variations were found.

The answer to **Question c** is also NO. Fig. S3 is not relevant to this topic. The figure regarding the two conformations of the SLP/core $\sigma 2$ proteins is to demonstrate that $\sigma 2$ binds to the inner shell protein $\lambda 1$ in three different configurations in each MRV particle: $\sigma 2_1$ and $\sigma 2_2$ adopt a uniform configuration in the asymmetric unit, and $\sigma 2_3$ occupies two equally distributed opposite directions, shared by two asymmetric units, leading to a total of 150 $\sigma 2$ copies in one MRV particle ($\sigma 2_1 \times 60 + \sigma 2_2 \times 60 + \sigma 2_3 \times 60/2$). See lines 416-425 for details.

Reviewer #2 (Remarks to the Author):

The manuscript by Liu et al. investigates the architecture and dynamics of several mammalian reovirus (MRV) assembly intermediates at both early and late stages post-infection using a combination of cryo-electron tomography (cryo-ET) on cryo-focused ion beam (cryo-FIB) SEM lamellae and cryo-electron microscopy (cryoEM). This study elucidates the molecular interactions between these intermediates and their interplay with

cellular components, as well as the temporal sequence and spatial rearrangement among them. The manuscript presents a significant piece of high-quality structural work with a well-described experimental approach that is both sound and robust. The data have been rigorously analyzed, and the paper is well-written. The results contribute significantly not only to the field of MRV research but also to the broader understanding of reoviridae assembly. The manuscript merits publication in its current form with a few revisions:

Ans: Thank you for your support! Your comments are addressed below.

1. Page 6: The presentation of different single-layered particles in the first paragraph (line 102: star core, core, and empty core) and the second paragraph (line 109: star core, hexagonal core, and spherical core (referred to as core hereafter)) is confusing. This terminology should be clarified and homogenized. Additionally, the hexagonal core is introduced at this point in the manuscript but is not mentioned afterward. It is unclear how it "disappears," and this should be explained.

Ans: This has been clarified in the revised manuscript. We retained the terminology of "star core" and "core" and used "hexagon-shaped SLP" instead of "hexagonal core" in the revised manuscript. See lines 124. As described in lines 128-130, during genome packaging and RNA replication, the capsid of SLP propagates from a star-shaped to a hexagon-shaped and then to a spherical particle. The hexagon-shaped SLP, identified in tomograms with a total of approximately 15 particles, did not yield a reconstruction, and could not be sorted from the extract of infected cells, indicating that this is a transient and unstable state. Therefore, we did not introduce the terminology "hexagonal core" in the revised manuscript.

2. Page 6, Line 111: The sentence "Both the star core and hexagonal core contain internal densities, though only the spherical ones are fully occupied" requires clarification. The specific internal densities being referred to and what is meant by "fully occupied" should be explained in more detail.

Ans: This has been addressed in the revised manuscript. The internal density within spherical SLP appears identical to that of the virion particle, indicating that dsRNA is fully packaged in the spherical SLP. In contrast, the densities inside star-shaped and hexagon-shaped SLPs are much lower than the dsRNA densities visualized in virion particles, suggesting that only ssRNA or no RNA is present inside. Thus, replication from ssRNA to dsRNA occurs during the transition from star shape and hexagon shape to spherical shape, along with the capsid propagation at the 5-fold axis. The detailed explanation has been added to the revised manuscript. See lines 125-128.

3. Page 20, Lines 447-456: Data deposition codes are not specified. Additionally, all maps and PDBs described in the manuscript must be deposited in the database. For example:

- o Figure 1 presents six intermediate states of MRV assembly captured by cellular tomography, but only the star core (EMD-XXXXX) and virion (EMD-XXXXX) from subtomogram averaging of the dataset at 48 hours post-infection are deposited.

- o According to the manuscript, cryoEM data from 6 hours post-infection reveal the reconstruction of core, virion, empty core, and empty virion (Figures 3 and S3), and at 48 hours post-infection, structures of the virion, empty virion, and outer shell particle were obtained (Figure S4). However, only the core (EMD-XXXXX), unturreted empty core (EMD-XXXXX), turreted empty core (EMD-XXXXX), and virion (EMD-XXXXX) from the dataset at 6 hours post-infection; and empty virion (EMD-450XXXXX) and outer shell (EMD-XXXXX) from the 48-hour post-infection dataset have been deposited.

Ans: All maps and PDBs described in the manuscript have been deposited in the database with codes listed in Data and code availability. See lines 467-476 in Page 20.

4. Figure S2: Equivalent particles at different time points appear significantly different (e.g., 6 hpi virion vs. 48 hpi virion). It is unclear whether these differences are genuine and should be addressed in the manuscript, or if they are due to differences in representation (e.g., the color scale used in each 3D rendering). This should be clarified.

Ans: This has been clarified in the figure legend. In Figure S2, the viral particles are colored by radius using consistent parameter settings. For the empty core and filled core particles from the two time points, the occupancy of the turret and $\sigma 2$ proteins differs. Equivalent particles at different time points were reconstructed independently from distinct particle sets. Therefore, the resulting maps have different resolutions and signal-to-noise levels, which make the particles appear different (virion, empty virion and the outer shell). Figure legend has been modified to make this clear: "Icosahedral reconstructions are colored by radius in a and b."

Reviewer #3 (Remarks to the Author):

The paper presents a thorough structural examination of mammalian reovirus assembly intermediates visualized in cells isolated from virus-infected cells. These studies add more exciting data to the existing recently published literature, where similar studies of virus replicative intermediates characterized by FIB milling, cryo-ET, and SRP have been done with other members of the Reoviridae family, including BTV, rotaviruses, and reoviruses. I enjoyed reading the section on RNA incorporation at the 5-fold axes. I felt these assembly details should be the focus of the paper rather than the cell-virus interactions only touched on in the first section. However, my main criticism concerns that the authors make a point of studying the condensate without showing any data that reovirus replicative organelles are formed through LLPS, or present a condensate. The title describes “molecular sociology” to describe cell-virus interactions. This should be more clearly stated in the title. Most of the content is on MRV assembly stages and packaging (‘choreography’ as they neatly put it), which I think would make for a more representative title. Overall, the manuscript reads as another nice piece of work that addresses the visualization of viral intermediates in cells using cryo-ET, and it would be misleading to state that it sheds light on how the viral condensates are formed, or how they support viral replication, nor they reveal such dynamics of the interactions within the condensates, or with other cellular organelles (‘sociology’).

Ans: Thank you for your generous support of our work! Your points concerning the terminologies of condensates and molecular sociology are well taken and are retained in the revised manuscript for the reasons detailed below.

Broader points:

1. -Terminology and wording surrounding condensates were a bit loose throughout the text, including evidence.

(Question a) They should give their definition of a condensate-membrane-less inclusion.

(Question b) Is LLPS a requirement, or are other processes that form inclusions also capable of making condensates?

(Question c) Size requirement?

(Question d) Liquid-like only, what about changes in properties over time/as viral arrays form?

Ans: For **Question a**, we added the definition of cellular condensates in introduction with references, see lines 27-31. “Cellular condensates are molecule-dense regions distinctive from their surroundings in a cell without the use of membrane. Some are also referred to as liquid-like phases, granules, and in the case of viral replication or assembly in host cells, viral factories. Such condensates selectively enrich specific molecules and facilitate biological reactions (Banani et al., Nat Rev Mol Cell Biol, 2017; Lee et al., mBio, 2021)^{3,4}.”

For **Question b**, LLPS is a major mechanism for forming condensates, while other processes can also lead to the formation of condensates (Lyon et al., Nat Rev Mol Cell Biol, 2021; Boeynaems et al., Trends Cell Biol, 2018)^{5,6}, such as gelation or formation of amyloid-like fibers (Boke et al., Cell, 2016)⁷.

For **Question c**, The size of cellular condensates range from micrometer-sized bodies such as the nucleolus and stress granules, to submicrometer structures, being <100-300 nm in diameter, such as transcriptional assemblies and some signaling punctas (Lyon et al., Nat Rev Mol Cell Biol, 2021; Shin et al., Science, 2017)^{5,8}. The size range seems to be broad.

For **Question d**, Many biomolecular condensates possess liquid-like properties. However, some appear to behave more like solids (Kroschwald et al., Elife, 2015)⁹. Moreover, physical properties of phase-separated droplets can change over time, initially fluid and eventually behaving as solid. This process is referred to as maturation or hardening (Banani et al., Nat Rev Mol Cell Biol, 2017)³. This phenomenon was also reported in rotavirus-induced cellular condensates. They show that at later infection (> 8– 12 h) stage, condensates/viropasms undergo a liquid-to-solid transition (Geiger et al., EMBO J, 2021)¹⁰.

2. Evidence of MRV liquid-like properties (still not definitively LLPS) is not mentioned. PMC8406312 mentions LLPS of the MRV μ NS protein, although more work needs to be done to address this question, as MRV replicative organelles have not been formally described as ‘condensates’ in literature yet.

Ans: More evidence including the suggested reference has been incorporated in the revised manuscript. One sentence “The nonstructural protein μ NS forms the scaffold of viral factories, which can be considered as dynamic liquid-like condensates.” with references (Lee et al., mBio, 2021; Bussiere et al., J Virol, 2017; Barkley et al., Mol Biol Cell, 2024; Broering et al., J Virol, 2002; Miller et al., J Virol, 2010) has been added. See lines 59-60.

The **evidence of MRV liquid-like properties** is shown in references (Lee et al., mBio, 2021; Bussiere et al., J Virol, 2017; Barkley et al., Mol Biol Cell, 2024;)^{4,11,12}, including the reference (PMC8406312) mentioned by the reviewer.

Regarding to the use of the trendy terminology of “cellular condensates”, we respectfully disagree with the reviewer. Indeed, several published studies **have already used the term “condensates” in the context of MRV**. For example:

- The μ NS protein forms the scaffold for reovirus factories (Broering et al., J Virol, 2002; Miller et al., J Virol, 2010)^{13,14} that resemble liquid-liquid phase-separated condensates (Lee et al., mBio, 2021)⁴. It is stated in PMC8406312 (Lee et al., mBio, 2021) that “it is likely that μ NS does not require RNA to phase separate”; and that “Future studies will identify the minimal constituents and conditions required to form **reovirus factory condensates**.”⁴
- A paper published by John Parker’s group in June 2024 (Barkley et al., Mol Biol Cell, 2024)¹², clearly states “The findings deepen our understanding of mammalian orthoreovirus viral factory formation by defining the nonstructural reoviral protein μ NS as a phase-separating biomolecule that nucleates **dynamic liquid-like condensates**.” It further states, “This study shows that μ NS, the viral nonstructural protein that forms ReoV viral factories, self-assembles into **phase-separated compartments with dynamic liquid-like properties**.” Additionally, the paper highlights, “**ReoV viral factories** have shown properties consistent with **biomolecular condensates** (Bussiere et al., 2017). **This work adds ReoV to the growing list of viruses that form phase-separated compartments.**” This paper is cited in the revised manuscript (line 60).
- All the evidence listed above demonstrates the liquid-like properties of MRV viral factories and further verifies that μ NS is a phase-separating biomolecule that nucleates dynamic liquid-like condensates. Thus, MRV viral factories are biomolecular condensates formed by LLPS of μ NS.

Nonetheless, we respect the reviewer’s concern, and, in the revised manuscript, we now clearly define cellular condensates and conventional term of viral factories (see line 27-31) before we used the term.

3. -(Qa) Important literature on MRV not touched on, including related evidence of LLPS. **(Qb)** Lipid droplets or cellular membranes incorporated in condensates are not expanded on.

For example:

membrane plays important role in the replication factory of MRV: Tenorio et al show μ NS and sigmaNS remodel ER membrane for interaction with inclusion bodies

doi: 10.1128/mBio.01253-18; or 10.1371/journal.ppat.1010641 These important papers are not discussed and may go against their condensate definition.

Ans: Your suggestion has been followed.

(Qa) The related evidence of LLPS has been incorporated into the revised manuscript. See the detailed description and explanation in the Ans to Q2.

(Qb) Lipid droplets and cellular membranes have been discussed in the revised manuscript as suggested. The sentences “Endoplasmic reticulum membranes were reported to be remodeled by reovirus non-structural protein σ NS or μ NS into tubular or vesicular forms, participating in the formation of MRV viral factories/condensates (doi: 10.1128/mBio.01253-18)¹⁵. Lipid droplets have been reported to play a role in the regulation and assembly of outer capsid proteins (10.1371/journal.ppat.1010641)².” have been added. See lines 101-104.

The different methods used in these studies, such as fluorescence microscopy and biochemical analysis, complement our results and provide additional information for our findings.

4. Figs 1 & S1- viral factories are not spherical, despite sphericity being used as an argument for LLPS in the introduction. This should be discussed, and some correlative imaging of condensates should be added to strengthen their argument that these are indeed condensates.

Ans: Cellular condensates that represent phase-separated liquid states do not always appear as homogeneous spherical droplets. Instead, they can be non-spherical shapes and exhibit some degree of internal structuring (Shin et al., Science, 2017)⁸. One of the most well-known examples is the nucleolus, which exhibits a core-shell architecture (Feric et al., Cell, 2018)¹⁶. Stress granules and P bodies often appear irregular and asymmetrical rather than consistently spherical (Sheth et al., Development, 2010; Wheeler et al., Elife, 2016)^{17,18}.

Studies on fluorescence microscopy of the MRV-induced cellular condensates show that they are irregular and not always spherical (Kniert et al., PLoS Pathog., 2022; Bussiere et al., J Virol, 2017; Barkley et al., Mol Biol Cell, 2024; Miller et al., J Virol, 2003)^{2,11,12,19}.

5. Figure 6 – **(Qa)** word coloring hard to read (blue of μ 1 and yellow of viral factory). **(Qb)** Condensate elongated shape may be misleading. **(Qc)** What is the zoomed-in section on the bottom left referring to?

Ans: (Qa) Fixed. We modified the color scheme of μ 1 and viral factory to facilitate reading.

(Qb) Fixed. We have clarified in the figure legend that: Depictions are not drawn to scale; the elongated shape of the viral factory/condensate does not reflect its actual shape, which is pleomorphic.

(Qc) The zoomed-in inset at the bottom left corner depicts μ NS interactions that are involved in forming MRV-induced condensates. This has been clarified in the figure legend of Fig. 6.

By line:

6. 2- not all of the replication cycle necessarily takes place in factories.

-not all viral factories are condensates, e.g. membrane-bound replication organelles.

-Narrowing down to particular taxa or viral genome type (e.g. dsRNA) may be needed.

Ans: We fully agree with you that not all viral replication cycles take place in factories, nor are all viral factories considered condensates. We changed "where viral replication occurs" to "where replication of many viruses occurs" to be more accurate. See line 3.

This statement can't be restricted to specific taxa or viral genome type. For example, rabies virus, vesicular stomatitis virus (VSV), and measles virus are **ssRNA** virus, while rotavirus and MRV are **dsRNA** virus; the replication cycles of these viruses take place in viral factories that are cellular condensates. Additionally, a study by Caragliano et al. (Viruses, 2022) shows that biomolecular condensates form during both herpesvirus (**dsDNA virus**) latency and lytic replication²⁰.

7. 8- should change "dsRNA virus" to MRV specifically, same rules/mechanisms don't apply for all.

Ans: Fixed. Modified as suggested. See line 7.

8. 10: condensate not proven to "enable" replication and assembly, more studies showing impairment with disrupted MRV condensate are needed.

Ans: We changed "enables" to "supports" in the revised manuscript. See line 10.

Two independent studies from Terence Dermody & James D. Chappell's group and Max Nibert's group show that MRV viral factories/condensates support viral replication^{21,22}. μ NS mutants specifically defective in forming factory-matrix structures, leading to the diffuse distribution of μ NS in cells, prevented the rescue of viral growth.

9. 28: **(Qa)** The definition of condensate needs improving. **(Qb)** Condensates are not phase separation (= a process), nor a granule. These terms are not interchangeable

Ans: (Qa) We added the definition of cellular condensates in introduction with references, see Ans to Q1a.

(Qb) Fixed. We deleted "also known as phase separation or granules," in the revised manuscript. See line 31.

10. 32: cellular condensates not indispensable for all viral lifecycles. e.g. rotavirus can still replicate, just impaired

Ans: Fixed. We changed "indispensable" to "important" and changed "in viral life cycles" to "in the life cycles of many viruses" in the revised manuscript. See line 35-36.

11. 36: **(Qa)** The viral condensates referenced should be explicitly stated as LLPS properties don't apply to all. **(Qb)** Also, growth/fusion shows liquid-like but not definitively LLPS.

Ans: (Qa) Fixed. One sentence with references was added: "Growing evidence shows that liquid-liquid phase separation mediates the formation of certain viral condensates^{12,23}." See lines 40-42.

(Qb) Fixed. We changed "demonstrates their properties of liquid-liquid phase separation" to "demonstrates their liquid-like properties". See line 40.

12. 66: temporal feels like a bit of a stretch given they rely entirely in this section on 2 images taken at 6h and 48h. No dynamics can be understood, lots of things can happen between these points that are lost.

Ans: Fixed. We changed "their temporal changes in host cells at different time points post-infection" to "the changes in host cells at two time points post-infection". See lines 65-66.

13. 70: **(Qa)** ‘Clearly show that MRV VFs are non-membrane-bound condensates with an ambiguous boundary’ – does this mean it’s not as clear if the boundary is ambiguous? **(Qb)** Why is the shape not spherical if it’s supposed to be a spherical droplet?

Ans: **(Qa)** We deleted “with an ambiguous boundary”. See line 76.

(Qb) Please see Ans to Q4.

14. 73: Same principle, due to lack of temporal information there is no evidence of “merging” to form large condensate from smaller. Ostwald Ripening is also possible. Either way, these data only show static snapshots of small regions of interest in frozen cells – there’s no evidence of ‘merging’.

Ans: Fixed. We changed “these smaller condensates merge into one large condensate that appears as a paracrystalline array of viral particles” to “one large condensate appears as a para-crystalline array of viral particles”. See line 78. Two sentences with reference “Live-cell microscopy has revealed dynamic fusion events for MRV condensates in which small condensates merge with larger ones¹¹. The large condensate we observed at 48 h post-infection is likely the result of such fusion events.” were added to the revised manuscript. See lines 79-82.

15. 84. a statement that dynein ‘transports nascent virions’ – the reference provided refers to unrelated viruses (Rabies for Ref 28), and without additional live-cell imaging studies with decorated dyneins or their cargos, it’s impossible to tell what these densities are at this resolution. I would remove this speculative statement, or tone it down.

Ans: We agree with the reviewer that, at this resolution, it is difficult to determine the identity of the decoration. Eichwald et al. (Virology, 2018) reported that dynein is localized in MRV induced condensates and is required for the formation of large condensates²⁴. The manuscript has been modified to include this reference, and the reference Rabies has been removed. We changed the statement to “Dynein, a microtubule motor protein, has been localized in MRV induced condensates and shown to be required for the forming large condensates. Whether the microtubule-decorating densities (Fig. S1e) are dynein motors or not awaits further investigation.” See lines 89-92.

16. 89. ‘ribosomes in the vicinity of condensates’ – I would discuss this result in light of this work published by John Parker’s group: <https://doi.org/10.1128/mbio.01463>

Ans: The reviewer’s suggestion has been followed and we have revised the manuscript accordingly. One sentence with the reference has been added: “This observation aligns with earlier immunofluorescent results showing that active translation of reovirus mRNA occurs within MRV viral factories, and that the translational machinery, including ribosomal subunits and cellular translation factors, localizes to viral factories²⁵.” See lines 96-99.

17. 92. I don’t find any “sociology” between virus and cell in this section, just colocalization. No basis for the “hijacking” is also given despite being mentioned here - **(Qa)** another study used nocodazole to show microtubule association and importance previously, and this is given in reference 27 published over 20 years ago. **(Qb)** No data shown to support that nearby ribosomes are particularly responsible for viral protein synthesis, **(Qc)** or the role of cell membrane etc. in condensates.

Ans: **(Qa)** We added two more references to support the importance of microtubule to MRV condensates (Bussiere et al., J Virol, 2017; Eichwald et al., Virology, 2018)^{11,24}. See line 95.

(Qb) Discussion about published data has been added. See Ans to Q16.

(Qc) Discussion about published data has been added. See Ans to Q3b.

Our cryoET observations directly show the colocalization and interplay between host and viral components. Together with the above-mentioned functional results, all of which are incorporated in the revised manuscript (see line 104), this suggests that MRV hijacks host machinery for its replication. We hope the reviewer agrees that these findings collectively illustrate the “sociology” between virus and cell.

18. 144. Please drop the term ‘viral isolate’ as this term has a different meaning in virology, and it can be confusing.

Ans: Fixed. “the viral isolate” has been replaced by “extract of MRV-infected cells”. See lines 160-161.

19. 164: The authors should discuss how the RNA genome facilitates turret attachment in their model. Does this occur directly, or by changing shell shape/stability to enable attachment?

Ans: Please refer to the Ans to Q3 of reviewer #1. We added the hypothesis in the revised manuscript. See lines 201-203.

20. 205: The statement that assembled core particles in the condensate are active in mRNA transcription has two references (32, 33) that do not show anything about transcription within condensates. Both ref. 32 & 33 show that cores are transcriptionally active in vitro, and when the reovirus muNS protein is added to cores and sticks to them (Ref.33). Unless the authors have their own data to demonstrate that they can visualize transcription in situ inside viral factories, they should remove this statement.

Ans: Fixed. Several studies (Kniert et al., PLoS Pathog, 2022; Miller et al., J Virol, 2010) show the newly synthesized MRV RNA is localized within the condensates, suggesting that core particles within the condensates are active in mRNA transcription^{2,14}. These references have been added to the revised manuscript. See line 224.

21. 250: 'Cellular condensates are highly regulated structures rather than chaotic accumulations' – this is not what the paper shows, as many previous studies on condensates have already demonstrated that condensates are not aggregates. I would also use the term 'aggregate' instead of 'chaotic accumulation'.

Ans: Fixed. We narrowed down the statement to "MRV-induced" in the revised manuscript and replaced "chaotic accumulation" with "sporadic aggregates". See lines 268-269.

In **summary**, we are grateful to you for the careful reviews and the suggested improvements, which have been all incorporated in the revised manuscript and figures. Thank you again for time and input!

Reference:

- 1 Broering, T. J., McCutcheon, A. M., Centonze, V. E. & Nibert, M. L. Reovirus nonstructural protein muNS binds to core particles but does not inhibit their transcription and capping activities. *J Virol* **74**, 5516-5524 (2000). <https://doi.org/10.1128/jvi.74.12.5516-5524.2000>
- 2 Kniert, J. et al. Reovirus uses temporospatial compartmentalization to orchestrate core versus outer capsid assembly. *PLoS Pathog* **18**, e1010641 (2022). <https://doi.org/10.1371/journal.ppat.1010641>
- 3 Banani, S. F., Lee, H. O., Hyman, A. A. & Rosen, M. K. Biomolecular condensates: organizers of cellular biochemistry. *Nat Rev Mol Cell Biol* **18**, 285-298 (2017). <https://doi.org/10.1038/nrm.2017.7>
- 4 Lee, C. H. et al. Reovirus Nonstructural Protein sigmaNS Recruits Viral RNA to Replication Organelles. *mBio* **12**, e0140821 (2021). <https://doi.org/10.1128/mBio.01408-21>
- 5 Lyon, A. S., Peeples, W. B. & Rosen, M. K. A framework for understanding the functions of biomolecular condensates across scales. *Nat Rev Mol Cell Biol* **22**, 215-235 (2021). <https://doi.org/10.1038/s41580-020-00303-z>
- 6 Boeynaems, S. et al. Protein Phase Separation: A New Phase in Cell Biology. *Trends Cell Biol* **28**, 420-435 (2018). <https://doi.org/10.1016/j.tcb.2018.02.004>
- 7 Boke, E. et al. Amyloid-like Self-Assembly of a Cellular Compartment. *Cell* **166**, 637-650 (2016). <https://doi.org/10.1016/j.cell.2016.06.051>
- 8 Shin, Y. & Brangwynne, C. P. Liquid phase condensation in cell physiology and disease. *Science* **357** (2017). <https://doi.org/10.1126/science.aaf4382>
- 9 Kroschwald, S. et al. Promiscuous interactions and protein disaggregases determine the material state of stress-inducible RNP granules. *Elife* **4**, e06807 (2015). <https://doi.org/10.7554/eLife.06807>
- 10 Geiger, F. et al. Liquid-liquid phase separation underpins the formation of replication factories in rotaviruses. *EMBO J* **40**, e107711 (2021). <https://doi.org/10.15252/embj.2021107711>
- 11 Bussiere, L. D., Choudhury, P., Bellaire, B. & Miller, C. L. Characterization of a Replicating Mammalian Orthoreovirus with Tetracysteine-Tagged muNS for Live-Cell Visualization of Viral Factories. *J Virol* **91** (2017). <https://doi.org/10.1128/JVI.01371-17>
- 12 Barkley, R. J. R., Crowley, J. C., Brodrick, A. J., Zipfel, W. R. & Parker, J. S. L. Fluorescent protein tags affect the condensation properties of a phase-separating viral protein. *Mol Biol Cell* **35**, ar100 (2024). <https://doi.org/10.1091/mbc.E24-01-0013>
- 13 Broering, T. J., Parker, J. S., Joyce, P. L., Kim, J. & Nibert, M. L. Mammalian reovirus nonstructural protein microNS forms large inclusions and colocalizes with reovirus microtubule-associated protein micro2 in transfected cells. *J Virol* **76**, 8285-8297 (2002). <https://doi.org/10.1128/jvi.76.16.8285-8297.2002>
- 14 Miller, C. L., Arnold, M. M., Broering, T. J., Hastings, C. E. & Nibert, M. L. Localization of mammalian orthoreovirus proteins to cytoplasmic factory-like structures via nonoverlapping regions of microNS. *J Virol* **84**, 867-882 (2010). <https://doi.org/10.1128/JVI.01571-09>

- 15 Tenorio, R. *et al.* Reovirus sigmaNS and muNS Proteins Remodel the Endoplasmic Reticulum to Build Replication Neo-Organelles. *mBio* **9** (2018). <https://doi.org/10.1128/mBio.01253-18>
- 16 Feric, M. *et al.* Coexisting Liquid Phases Underlie Nucleolar Subcompartments. *Cell* **165**, 1686-1697 (2016). <https://doi.org/10.1016/j.cell.2016.04.047>
- 17 Sheth, U., Pitt, J., Dennis, S. & Priess, J. R. Perinuclear P granules are the principal sites of mRNA export in adult *C. elegans* germ cells. *Development* **137**, 1305-1314 (2010). <https://doi.org/10.1242/dev.044255>
- 18 Wheeler, J. R., Matheny, T., Jain, S., Abrisch, R. & Parker, R. Distinct stages in stress granule assembly and disassembly. *Elife* **5** (2016). <https://doi.org/10.7554/eLife.18413>
- 19 Miller, C. L., Broering, T. J., Parker, J. S., Arnold, M. M. & Nibert, M. L. Reovirus sigma NS protein localizes to inclusions through an association requiring the mu NS amino terminus. *J Virol* **77**, 4566-4576 (2003). <https://doi.org/10.1128/jvi.77.8.4566-4576.2003>
- 20 Caragliano, E., Brune, W. & Bosse, J. B. Herpesvirus Replication Compartments: Dynamic Biomolecular Condensates? *Viruses* **14** (2022). <https://doi.org/10.3390/v14050960>
- 21 Arnold, M. M., Murray, K. E. & Nibert, M. L. Formation of the factory matrix is an important, though not a sufficient function of nonstructural protein mu NS during reovirus infection. *Virology* **375**, 412-423 (2008). <https://doi.org/10.1016/j.virol.2008.02.024>
- 22 Kobayashi, T., Ooms, L. S., Chappell, J. D. & Dermody, T. S. Identification of functional domains in reovirus replication proteins muNS and mu2. *J Virol* **83**, 2892-2906 (2009). <https://doi.org/10.1128/JVI.01495-08>
- 23 Etibor, T. A., Yamauchi, Y. & Amorim, M. J. Liquid Biomolecular Condensates and Viral Lifecycles: Review and Perspectives. *Viruses* **13** (2021). <https://doi.org/10.3390/v13030366>
- 24 Eichwald, C., Ackermann, M. & Nibert, M. L. The dynamics of both filamentous and globular mammalian reovirus viral factories rely on the microtubule network. *Virology* **518**, 77-86 (2018). <https://doi.org/10.1016/j.virol.2018.02.009>
- 25 Desmet, E. A., Anguish, L. J. & Parker, J. S. Virus-mediated compartmentalization of the host translational machinery. *mBio* **5**, e01463-01414 (2014). <https://doi.org/10.1128/mBio.01463-14>